# Disruption of auto-inhibition underlies conformational signaling of ASIC1a to induce neuronal necroptosis

Jing-Jing Wang[1,4], Fan Liu[1,4], Fan Yang[2], Yi-Zhi Wang[1], Xin Qi[1], Ying Li[1], Qin Hu [1*], Michael X. Zhu [3*] & Tian-Le Xu[1*]

We reported previously that acid-sensing ion channel 1a (ASIC1a) mediates acidic neuronal necroptosis via recruiting receptor-interacting protein kinase 1 (RIPK1) to its C terminus (CT), independent of its ion-conducting function. Here we show that the N-terminus (NT) of ASIC1a interacts with its CT to form an auto-inhibition that prevents RIPK1 recruitment/ activation under resting conditions. The interaction involves glutamate residues at distal NT and is disrupted by acidosis. Expression of mutant ASIC1a bearing truncation or glutamate-to-alanine substitutions at distal NT causes constitutive cell death. The NT-CT interaction is further disrupted by N-ethylmaleimide-sensitive fusion ATPase (NSF), which associates with ASIC1a-NT under acidosis, facilitating RIPK1 interaction with ASIC1a-CT. Importantly, a membrane-penetrating synthetic peptide representing the distal 20 ASIC1a NT residues, $NT_{1-20}$, reduced neuronal damage in both in vitro model of acidotoxicity and in vivo mouse model of ischemic stroke, demonstrating the therapeutic potential of targeting the auto-inhibition of ASIC1a for neuroprotection against acidotoxicity.

[1] Collaborative Innovation Center for Brain Science, Department of Anatomy and Physiology, Shanghai Jiao Tong University School of Medicine, Shanghai, China. [2] Department of Biophysics and Kidney Disease Center, First Affiliated Hospital, Institute of Neuroscience, National Health Commission and Chinese Academy of Medical Sciences Key Laboratory of Medical Neurobiology, Zhejiang University School of Medicine, Hangzhou, China. [3] Department of Integrative Biology and Pharmacology, McGovern Medical School, The University of Texas Health Science Center at Houston, Houston, USA. [4]These authors contributed equally: Jing-Jing Wang, Fan Liu. *email: huqinle20010709@126.com; michael.x.zhu@uth.tmc.edu; xu-happiness@shsmu.edu.cn

Tissue acidosis is well-recognized as one of the major contributors to neuronal cell death in many neurological diseases. An improved understanding of signaling pathways mediating acidotoxicity, therefore, will help devise better therapeutic strategies for neuroprotection. Among the various proton sensing mechanisms, those mediated by acid-sensing ion channels (ASICs), particularly subtype ASIC1a, are pivotal to acidosis-induced damage towards brain neurons[1,2]. The essential role of ASIC1a in neuronal acidotoxicity has been clearly demonstrated by the protective effect of ASIC1a gene deletion and/or pharmacological blockade in animal models of ischemic stroke[1], multiple sclerosis[3], Huntington disease[4], Parkinson's disease[5], and spinal cord injury[6]. However, exactly how ASIC1a mediates neuronal cell damage has not been clearly illustrated.

Upon decrease in extracellular pH, ASIC1a is activated to produce a transient inward current mediated mainly by $Na^+$, with a small component of $Ca^{2+}$ [1,7–9]. Traditionally, the $Ca^{2+}$ influx is thought to cause cytosolic $Ca^{2+}$ overload and thereby acidotoxicity[1,7–9]. However, we recently uncovered that the ion conducting function of ASIC1a is not required for the extracellular acid-induced neuronal cell demise[2]. Rather, this acidotoxic effect involves an acidosis-evoked complex formation between ASIC1a and receptor-interacting serine/threonine-protein kinase 1 (RIPK1, also known as RIP1), with the latter becoming phosphorylated, eventually leading to a specific type of necrotic cell death, termed necroptosis[2]. Although not very common, ion-conduction independent conformational signaling has been reported to play important roles in some other ion channels, such as N-methyl-D-aspartate (NMDA) receptors, Kv10.1 $K^+$ channel, and Cav1.2 $Ca^{2+}$ channel[10–12]. This highlights the versatility of membrane protein signaling, where both channel and non-channel functions may be carried out by the same protein, and often triggered by the same stimulus; however, the downstream consequences can be dramatically different. Remarkably, the molecular details on how a channel executes between its ionotropic function and conformational signaling remain mysterious.

Here, we expand our previous work on conformational signaling of ASIC1a by revealing an auto-inhibition mechanism that regulates interactions between the N-terminus (NT) and C-terminus (CT) of ASIC1a protein. Disruption of this auto-inhibition underlies conformational signaling of ASIC1a to induce neuronal necroptosis. Testing synthetic membrane-penetrating peptides representing the distal NT of ASIC1a, we identify peptide $NT_{1–20}$ to offer protection to neurons in both acidosis-induced necroptosis in vitro and a mouse model of ischemic stroke in vivo, providing proof of concept for a neuroprotective strategy against acidotoxicity.

## Results

**Peptide $NT_{1–20}$ protects neurons against acidotoxicity.** Previously, we showed that ASIC1a-CT is critical for acidosis-induced cell death, which involves binding and phosphorylating RIPK1. A membrane-penetrating synthetic peptide containing the proximal CT region of mouse ASIC1a, CP-1, also induced RIPK1 phosphorylation and cell death even at neutral pH in both primary cultures of mouse cortical neurons and Chinese hamster ovary (CHO) cells[2]. To define the minimal amino acid sequence of this motif, we systematically omitted residues from either ends of CP-1 in the synthetic peptides (Supplementary Fig. 1a) and tested their ability to elicit death of mouse cortical neurons at neutral pH. The membrane-penetrating TAT peptide was tagged at the N-terminal end to allow transportation across plasma membrane. Using Cell Titer Blue (CTB) assay to assess cell viability, we found the nine residues

($K^{468}$CQKEAKRN$^{476}$, CP-1-2) in the middle of CP-1 to confer necroptosis (Fig. 1a, b; Supplementary Fig. 1a). The cytotoxic effect of CP-1-2 was also confirmed using lactate dehydrogenase (LDH) assay (Fig. 1c). The ability of CP-1-2 to elicit neuronal death was strongly reduced by further removal of three residues from its NT (CP-1-3) and abolished by omitting the TAT leading peptide (CP-1-2S) (Fig. 1b; Supplementary Fig. 1a). The latter effect suggests that membrane penetration is important for the cytotoxic effect of CP-1-2, consistent with its presumed cytoplasmic side of action.

Despite the presence of CP-1-2 sequence at ASIC1a proximal CT, the full length ASIC1a does not cause cell demise under normal physiological conditions, indicating that this death motif may be masked in the full-length channel at rest and only becomes exposed upon acid stimulation. As such, a binding partner of CP-1-2 might exist and be able to suppress its cytotoxic effect. The ASIC1a protein has two transmembrane helices, with both NT and CT located at cytoplasmic side[13,14]. We reasoned that because of the same side location and close proximity, ASIC1a-NT might serve the role of masking the death motif at ASIC1a-CT. To test this possibility, we first examined the ability of TAT-tagged peptides, $NT_{1–20}$, $NT_{11–30}$ and $NT_{21–41}$ (Supplementary Fig. 1b), representing various regions of ASIC1a-NT to suppress neuronal death induced by CP-1-2. Interestingly, co-application of $NT_{1–20}$, but not $NT_{11–30}$ or $NT_{21–41}$, markedly decreased the death of cortical neurons induced by CP-1-2 (Fig. 1c). Furthermore, the pretreatment with $NT_{1–20}$ also diminished cortical neuron death induced by exposure to pH 6.0 extracellular solution (Fig. 1d, e), supporting that ASIC1a-NT contains a functional motif capable of interacting with ASIC1a-CT death motif to prevent necroptotic damage. The protective effect of $NT_{1–20}$ was abolished by splitting the peptide into two halves, $NT_{1–10}$ and $NT_{11–20}$, (Fig. 1e). Also, the protection was detectable at 5 and 10 μM $NT_{1–20}$, but not 20 μM (Fig. 1f). Most likely, the loss of protection at 20 μM resulted from toxicity of the TAT tag, which has been known to induce inflammatory responses[15,16]. Indeed, 20 μM control (Ctrl) TAT peptide exacerbated the acid-induced neuronal death (Supplementary Fig. 1c). To avoid this complication, we used 10 μM $NT_{1–20}$ in all subsequent in vitro experiments.

As an alternative method to assess cell damage, we used Propidium Iodide (PI) to stain necrotic cells and showed that $NT_{1–20}$ (10 μM, added 0.5 h before application of the pH 6.0 solution) significantly reduced the number of PI-positive neurons (Fig. 1g, h). Importantly, $NT_{1–20}$ had no effect on acid (pH 6.0)-induced currents in cortical neurons (Fig. 1i, peak current amplitude with $NT_{1–20}$ was 93.7 ± 6.6% of that with Ctrl peptide, $n = 4$, $p > 0.05$, paired Student's $t$ test), supporting that the neuroprotection of $NT_{1–20}$ is independent of ionotropic function of ASIC1a. As negative controls, the peptide protected neither ASIC1a knockout (KO) neurons against acidotoxicity nor wild type (WT) neurons against excitotoxicity induced by glutamate (Supplementary Fig. 1d, e), indicating the specificity of $NT_{1–20}$ at ASIC1a-mediated acidotoxicity.

**Acidosis induces separation of ASIC1a-NT from ASIC1a-CT.** Because the cytoplasmic termini of ASIC1a were truncated in the high-resolution structures[13,14,17], we modeled ASIC1a full-length containing NT and CT de novo using the Rosetta suite[18] based on published closed and open state structures of this channel (Fig. 2a). The models suggest that in closed state, the highly positively charged proximal CT is in close proximity with distal NT, where the abundant presence of negatively charged residues likely allows electrostatic interactions (Fig. 2a, see also later). However, in open state, ASIC1a NT and CT are separated,

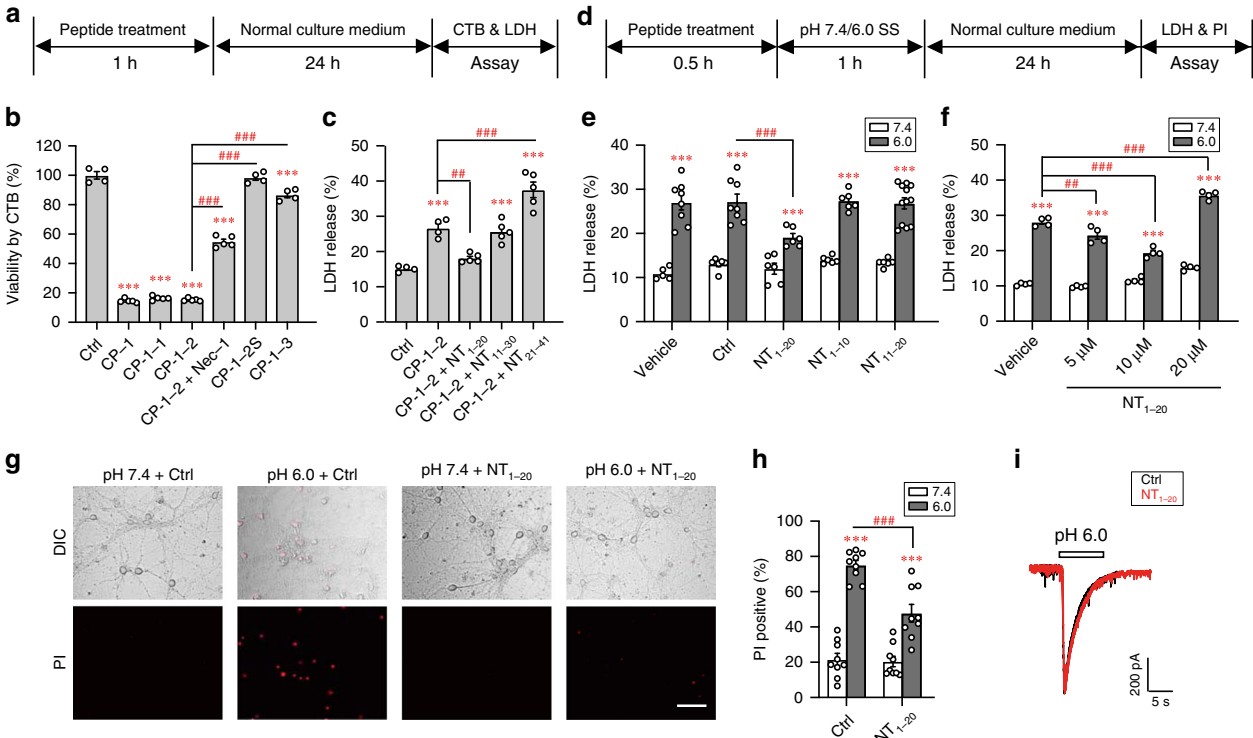

**Fig. 1 Peptide NT$_{1-20}$ protects mouse cortical neurons against acidosis-induced necroptosis. a** Schematics of cell death assays for (**b**, **c**). CTB, cell titer blue; LDH, lactate dehydrogenase. **b** CTB assay for viability of neurons treated with indicated peptides (all at 10 μM). CP-1, CP-1-1, CP-1-2 and CP-1-3 reduced viability. Nec-1 (20 μM) attenuated CP-1-2 induced death. Compared to CP-1-2, CP-1-3 induced much less and the membrane impermeable CP-1-2S induced no death. $n = 4$–6, *** $p < 0.001$ vs. control peptide (Ctrl, TAT alone), ### $p < 0.001$ vs. CP-1-2, by ANOVA. **c** LDH release assay for viability of neurons treated with CP-1-2 (10 μM) without or with NT peptides (10 μM). The toxicity by CP-1-2 was significantly reduced by NT$_{1-20}$, but not NT$_{11-30}$ and NT$_{21-41}$. $n = 4$–5, *** $p < 0.001$ vs. Ctrl, ## $p < 0.01$, ### $p < 0.001$ vs. CP-1-2, by ANOVA. **d** Schematics of cell death assay for (**e**–**h**). PI, propidium iodide (PI); SS, standard extracellular solution. **e** LDH release from neurons exposed to pH 7.4 and pH 6.0 SS. pH 6.0-induced death was reduced by treatment of 10 μM NT$_{1-20}$, but not NT$_{1-10}$ and NT$_{11-20}$. $n = 5$–12, *** $p < 0.001$ vs. corresponding pH 7.4, ### $p < 0.001$ vs. Ctrl in pH 6.0, by ANOVA. **f** Concentration dependence of NT$_{1-20}$ suppression of acidotoxicity. $n = 4$, *** $p < 0.001$ vs. corresponding pH 7.4, ## $p < 0.01$, ### $p < 0.001$ vs. vehicle in pH 6.0, by ANOVA. **g** DIC (upper) and PI staining (lower) images of neurons treated as in (**d**). Scale bar, 50 μm. **h** Summary for (**g**). $n > 200$ neurons counted for each. *** $p < 0.001$ vs. corresponding pH 7.4, ### $p < 0.001$ vs. Ctrl in pH 6.0, by ANOVA. **i** Pretreatment (0.5 h) with 10 μM NT$_{1-20}$ had no effects on pH 6.0-induced currents in cultured neurons. Representative current traces at −60 mV are shown. Error bars are SEM for all summary panels.

suggesting a gating-related conformational change that disrupts the N- to C-terminal interaction (Fig. 2a).

To confirm this prediction, we used Förster resonance energy transfer (FRET) technique to monitor relative distances between ASIC1a-NT and CT before and during the pH 6.0 treatment. A stable FRET signal was detected at neutral pH in CHO cells expressing human ASIC1a tagged with a cyan fluorescent protein (CFP) at NT and a yellow fluorescent protein (YFP) at CT (Supplementary Fig. 2a, b), consistent with the predicted intra-molecular N- to C-terminal interaction under resting conditions. Upon lowering extracellular pH to 6.0, the YFP/CFP emission ratio dropped slowly to reach a quasi-steady state after approximately 5 min. This change was reversible with wash-off of the pH 6.0 solution (Fig. 2b, c).

Using spectra FRET, we determined the energy transfer efficiency between the NT-tagged CFP and CT-tagged YFP on ASIC1a to be similar to that of the CFP-YFP concatemer (Fig. 2d; Supplementary Fig. 2c), indicating a very close interaction. Importantly, while acid treatment did not affect the FRET efficiency of CFP-YFP concatemer, it markedly decreased that of CFP-ASIC1a-YFP (WT) (Fig. 2d). Furthermore, the acid treat-ment also decreased FRET efficiency of a non-conducting mutant of ASIC1a that bears alanine substitutions at the three key residues, HIF, involved in ion permeation[19] (Fig. 2d, also CFP- and YFP-tagged). As the HIF mutant fails to generate any current

in response to acid but mediates acidosis-induced cell death[2], the acid-induced decrease in FRET efficiency likely reflects a conformational change at the cytoplasmic termini of ASIC1a relevant to cytotoxicity rather than ion conduction. This interpretation is in line with the slow kinetics of acid-evoked FRET changes seen in Fig. 2b, which is very different from the rapid development and then inactivation of acid-elicited current mediated by this channel, usually finishing in < 5 s (see Fig. 1i, for example).

On the other hand, with the conformational change of ASIC1a suppressed through disulfide bond formation by mutating Glu-235 and Tyr-389 to cysteines[20], CFP-ASIC1a-E235C/Y389C-YFP exhibited a slower and smaller acid-induced decrease in YFP/CFP ratio than WT (Fig. 2b, c). Moreover, the acid-induced decrease in YFP/CFP ratio of CFP-ASIC1a-YFP was significantly inhibited by treatment with the ASIC1a inhibitor, psalmotoxin-1 (PcTX1) (Fig. 2b, c). These results demonstrate a close relationship between acidosis-induced decrease in FRET signal of CFP-ASIC1a-YFP and conformational change of ASIC1a protein.

Intriguingly, although the ASIC1a-CT-tagged YFP (ASIC1a-YFP) is very sensitive to extracellular acidification, the presence of CFP (as in CFP-ASIC1a-YFP) strongly suppressed such a response (Supplementary Fig. 2d, e). YFP fluorescence is known to be quenched easily by protons. Because ASIC1a-CT tagging places YFP right underneath the plasma membrane, the rapid

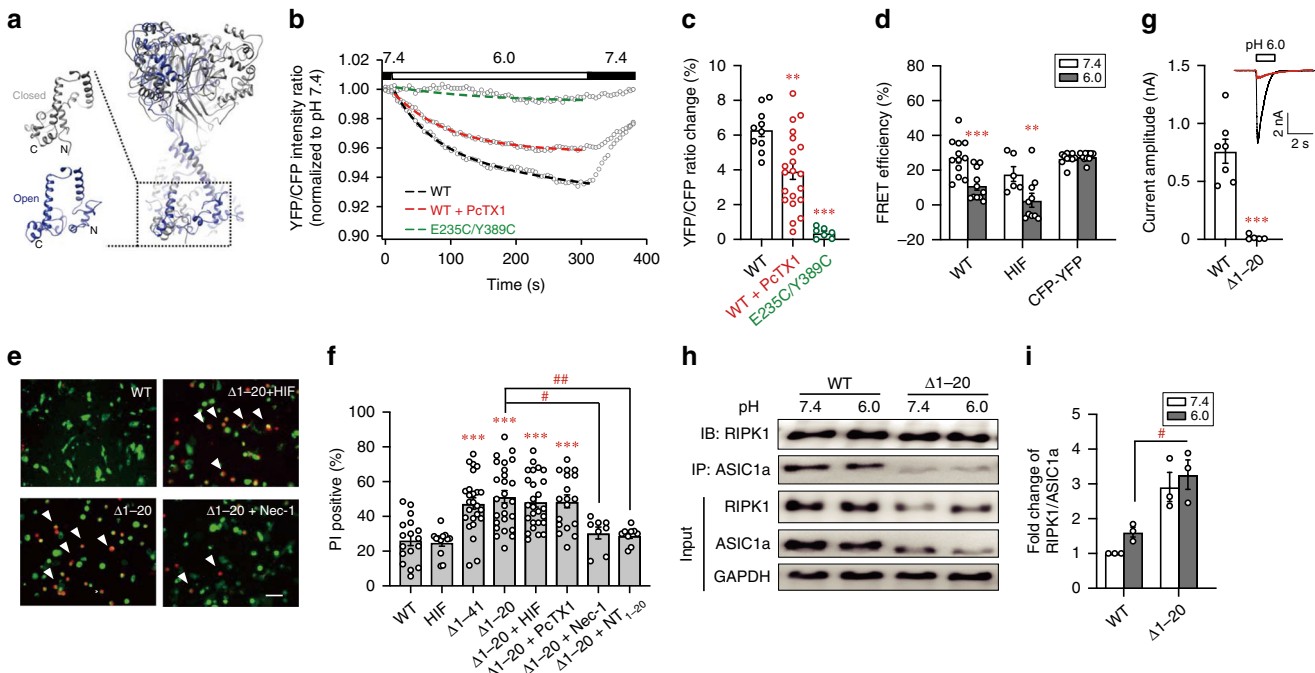

**Fig. 2 Distal N-terminal region of ASIC1a is crucial for inhibiting CP-1 death motif. a** De novo Rosetta modeling of full-length ASIC1a. Gray color represents closed state base on PDB structure 5wku, while blue color depicts open state based on PDB structure 4ntw. **b** Acidosis-induced dissociation of ASIC1a-CT from its NT, as measured by FRET: YFP/CFP emission ratio ($F_{525}/F_{482}$) with excitation at 405 nm. CHO cells expressing CFP-ASIC1a-YFP (WT) or CFP-ASIC1a-E235C/Y389C-YFP (E235C/Y389C) were untreated or treated with PcTX1 (100 nM). Bath solution pH changed from 7.4 to 6.0 as indicated. Data points are averages of $n = 10$-21 cells. Dashed lines are fits by exponential function. **c** Summary of % ratio changes at 5 min of pH 6.0 treatment for data in (**b**). ** $p < 0.01$, *** $p < 0.001$ *vs.* WT, by ANOVA. **d** Spectra FRET for energy transfer between CFP and YFP of WT, CFP-ASIC1a-HIF-YFP (HIF), and CFP-YFP at pH 7.4 and 6.0. $n = 6$-12, ** $p < 0.01$, *** $p < 0.001$ *vs.* corresponding pH 7.4, by ANOVA. **e, f** Representative images (**e**) and summary (**f**) of PI staining of CHO cells transfected with WT ASIC1a and its HIF and NT truncation mutants at pH 7.4. Scale bar, 20 μm. Nec-1 (20 μM), peptide $NT_{1-20}$ (10 μM), or PcTX1 (100 nM) was added immediately after transfection; PI staining was performed 24 h later. $n > 200$ cells counted for each. *** $p < 0.001$ *vs.* WT, # $p < 0.05$, ## $p < 0.01$ *vs.* Δ1-20, by ANOVA. **g** Δ1-20 (red trace) exhibited a large decrease in pH 6.0-evoked currents as compared to WT (black trace). Inset, representative current traces at −60 mV. Data are peak current amplitudes. $n = 5$-6, *** $p < 0.001$ *vs.* WT by unpaired t test. (**h**) Δ1-20 displayed decreased expression but increased association with RIPK1 at pH 7.4. Representative images of western blots for co-IP and inputs. **i** Summary data for RIPK1 pulled down by ASIC1a antibody. Data are normalized to RIPK1/ASIC1a of WT at pH 7.4. $n = 3$, # $p < 0.05$ *vs.* WT by ANOVA. Error bars are SEM for all summary panels.

decrease of ASIC1a-YFP fluorescence most likely reflected the near membrane cytoplasmic pH drop caused by extracellular acidification[2], which appeared to occur much faster than the global intracellular pH decrease measured by BCECF (Supplementary Fig. 3). The kinetics of YFP fluorescence change in CFP-ASIC1a-YFP in response to the pH 6.0 solution, however, more closely resembled the change in the YFP/CFP ratio (Supplementary Fig. 2e and Fig. 2b). The most plausible explanation for these observations would be that CFP in close vicinity prevented proton quenching of YFP fluorescence. Then the separation of CFP from YFP due to acid-evoked conformational change in ASIC1a caused decreases in not only energy transfer but also YFP fluorescence itself because of the increased proton quenching. Both of these would indicate structural changes at the cytoplasmic termini of ASIC1a. Notably, the near membrane cytoplasmic pH change, as shown by ASIC1a-YFP fluorescence drop (Supplementary Fig. 2e), was much faster than the conformational transition of ASIC1a, again indicating that structural changes at ASIC1a NT and CT are intrinsically much slower than the environmental pH changes both outside and inside the cell. Together, these data lend support to the hypothesis that acidotoxicity involves a conformational transition that disrupts the preformed N- to C-terminal interaction within ASIC1a to unmask the CT death motif, and this transition is kinetically distinct from environmental pH changes and channel gating, and it does not require the channel's ion conducting function.

**The distal NT of ASIC1a masks the death motif of its CT.** Given the ability of peptide $NT_{1-20}$ to suppress cytotoxicity of the CT death motif, we reasoned that truncating NT would free CT and result in constitutive activation of the death pathway. Indeed, when expressed in CHO cells, ASIC1a-NT truncation mutants, Δ1–41 or Δ1–20, led to increases in PI-positive cells compared to WT ASIC1a (Fig. 2e, f). The Δ1–20 deletion of the HIF mutant also caused a similar increase (Fig. 2e, f). The cytotoxic effect of Δ1–20 was unaffected by PcTX1, but attenuated by inhibiting RIPK1 with necrostatin-1 (Nec-1) or masking CT with $NT_{1-20}$ (Fig. 2e, f). Thus, cell death induced by the truncated ASIC1a is still mediated by RIPK1 and the unprotected CT death motif underlies the spontaneous killing.

Interestingly, whereas cells expressing WT ASIC1a developed large proton-evoked currents, those expressing the Δ1–20 mutant responded to pH 6.0 stimulation with very small currents (Fig. 2g). Similarly, the expression levels of Δ1–20 were much lower than WT ASIC1a (Fig. 2h). The reduced expression may be directly related to the ability of Δ1–20 to cause cell death, as cells with high levels of Δ1–20 expression would have been dead. Alternatively, ASIC1a might also undergo activity (or use)-dependent degradation, like RIPK2[2]. Consistent with the previous finding that ASIC1a-mediated RIPK1 activation also causes RIPK1 degradation[2], the levels of RIPK1 were also reduced in cells that expressed Δ1–20 (Fig. 2h). Despite the low level of expression, more RIPK1 was associated with Δ1–20 than WT

ASIC1a, as shown by co-immunoprecipitation (co-IP), and unlike WT, acidosis did not further increase RIPK1 association with the mutant ASIC1a (Fig. 2h, i). The lack of acid-induced increase in Δ1–20 association with RIPK1 could explain the failure of PcTX1 in suppressing cell death induced by this mutant, since proton stimulation was not required for the cytotoxic effect. Taken together, the above data strongly support our hypothesis that the distal NT serves a critical role in keeping the CT death motif of ASIC1a silent under normal physiological conditions, while acidosis induces a conformational transition to expose the CT death motif, which in turn recruits RIPK1 to trigger necroptosis.

**NSF binds to ASIC1a-NT to facilitate CT death motif exposure.** In parallel with the examination of N- to C-terminal interaction, we also searched for ASIC1a-associated proteins potentially involved in ASIC1a-mediated cell death, reasoning that protein-binding partners might impact ability of the CT death motif to activate RIPK1. To accomplish this, we performed immunoprecipitation using a polyclonal ASIC1a antibody against lysates from the contralateral (Contra) and ischemic hemispheres of the same cortical brain tissues of mice subject to intraluminal middle cerebral artery occlusion (MCAO), an experimental model of ischemic stroke that involves tissue acidosis and ASIC1a[21]. The immunoprecipitants were separated by SDS-PAGE and afterwards stained with Coomassie blue. Remarkably, a band with a molecular weight of ~85 kDa exhibited a clear increase in MCAO as compared to Contra samples (Fig. 3a). This band was collected and subject to matrix-assisted laser desorption/ionization time-of-flight mass spectrometry (MALDI-TOF-MS) analysis (Fig. 3b), which revealed 18 peptide sequences that matched N-ethylmaleimide-sensitive fusion ATPase (NSF), with the protein sequence coverage of 22% (Supplementary Data 1).

To confirm the interaction between NSF and ASIC1a, we performed co-IP using extracts from Contra and MCAO brain samples. The anti-ASIC1a antibody pulled down NSF from both Contra and MCAO groups, but the NSF levels were markedly higher in MCAO than Contra groups (Fig. 3c, upper panel, 3d). Consistent with the previous report[2], the association of RIPK1 with ASIC1a also increased after MCAO (Fig. 3c, upper panel, 3d). These data suggest that NSF and RIPK1 may also become associated in response to MCAO. Indeed, the anti-RIPK1 antibody pulled down more NSF from MCAO than Contra groups, along with markedly increased serine/threonine phosphorylation of RIPK1 (Fig. 3c, middle panel, 3d). Reciprocally, the anti-NSF antibody also pulled down more RIPK1 from MCAO than Contra brains (Fig. 3e, f). More remarkably, although there appeared to be basal association between NSF and RIPK1 in non-injured Contra brains, which was detectable in both WT and ASIC1a KO mice, the MCAO-induced increase was only observed in WT samples (Fig. 3e, f). This argues for a critical role of ASIC1a in bridging the association between NSF and RIPK1 in response to brain ischemia.

To verify if acidosis is responsible for the ASIC1a-mediated association between NSF and RIPK1, we performed co-IP on cultured mouse cortical neurons treated with pH 7.4 and pH 6.0 solutions for 30 min. The anti-NSF antibody pulled down RIPK1 in neurons exposed to pH 6.0 and this was strongly attenuated by PcTX1 added 30 min before the acid stimulation (Fig. 3g, h). These data suggest the formation of an NSF-ASIC1a-RIPK1 complex following ASIC1a activation by protons.

To define where NSF binds to ASIC1a, we employed the glutathione-S-transferase (GST) pull-down assay. GST fusion proteins containing ASIC1a-NT and ASIC1a-CT (referred to as GST-NT and GST-CT, respectively) were prepared and incubated with mouse brain lysates. Bound proteins were collected using glutathione agarose beads and subjected to immunoblotting by the anti-NSF antibody. Interestingly, only GST-NT, but not GST-CT or GST alone, pulled down NSF (Fig. 4a), suggesting that NSF is bound to ASIC1a-NT. To further probe NT regions critical for NSF binding, we expressed NT deletion mutants of ASIC1a (Fig. 4b) in CHO cells and performed co-IP after exposing the cells to pH 7.4 and pH 6.0 solutions. Interestingly, although there appeared to be some basal NSF-ASIC1a association at pH 7.4, which was unaffected by the deletions, the acid-induced increase, as seen for WT ASIC1a, was abolished in cells expressing Δ1–41 or Δ1–20 (Fig. 4c, d). Therefore, residues in aa1–20 likely contribute critically to ASIC1a interaction with NSF. Accordingly, a treatment with peptide $NT_{1-20}$, which would bind to both ASIC1a-CT and NSF, suppressed the acidosis-induced increase in ASIC1a association with not only RIPK1, along with less degradation of the RIPK1 protein, but also NSF, in cultured cortical neurons (Fig. 4e, f).

The neuroprotection by peptide $NT_{1-20}$ could result from its binding to ASIC1a-CT or NSF, or both. The binding to ASIC1a-CT would occlude the access of RIPK1 and thereby prevent necroptotic death; however, how would disrupting NSF affect acidosis-induced death? To answer this question, we first examined the effect of knocking down NSF by shRNA. Using an NSF-shRNA construct that also encodes GFP driven by the neuronal promoter Synp for identification of transfected neurons, we achieved effective knockdown of NSF expression (Fig. 5a, b). The treatment also reduced NSF-ASIC1a as well as RIPK1-ASIC1a interactions (Fig. 5c, d), without affecting cell surface expression of ASIC1a proteins and acid-evoked currents in cortical neurons (Supplementary Fig. 4a, b). In CHO cells expressing CFP-ASIC1a-YFP, the shRNA knockdown of NSF suppressed the acid-induced decrease in the YFP/CFP ratio (Fig. 5e, f), demonstrating the importance of NSF, most likely, in stabilizing the dissociation between ASIC1a-NT and CT under acidosis. Importantly, the shRNA knockdown of NSF attenuated acidosis-induced neuronal death, as shown by PI staining, in cultures prepared from WT mice, without affecting the viability of neurons prepared from ASIC1a KO mice (Fig. 5g, h). Moreover, N-ethylmaleimide (NEM), a nonspecific inhibitor of NSF, also protected neurons from acidotoxicity (Fig. 5i). These data indicate that NSF is specifically involved in acidosis-induced and ASIC1a-mediated neuronal death, i.e. conformational signaling of ASIC1a, but not its ion channel function.

**$E^6EEE^9$ in NT underlie ASIC1a intramolecular auto-inhibition.** As shown in Figs. 1e and 4e, f, peptide $NT_{1-20}$ attenuated not only acidosis-induced neuronal death but also the ASIC1a-NSF interaction. Given that ASIC1a distal NT is able to bind to both the CT death domain and NSF, both of these interactions may be competed for by the peptide to achieve neuroprotection. To better define the neuroprotective mechanism of peptide $NT_{1-20}$, we focused on the four consecutive glutamate residues ($E^6EEE^9$). In the Rosetta model of closed state ASIC1a, these negatively charged residues are juxtaposed to positively charged lysines ($K^{468}$, $K^{471}$, and $K^{474}$) in the CT death motif of the same subunit, with the distance between beta carbons of $E^7$ and $K^{468}$ estimated to be ~4.7 Å (Fig. 6a). Thus, the glutamate residues may be important for NT binding to its own ASIC1a-CT through electrostatic interactions. Therefore, disrupting $E^6EEE^9$ would more likely interfere with N- to C-terminal interaction than with ASIC1a-NSF association, since the latter requires residues in aa1–20 (Fig. 4c, d). Indeed, mutating all four glutamates into a-lanines (E/A mutant) resulted in decreased FRET (assessed by the absolute YFP/CFP ratios obtained under identical experimental

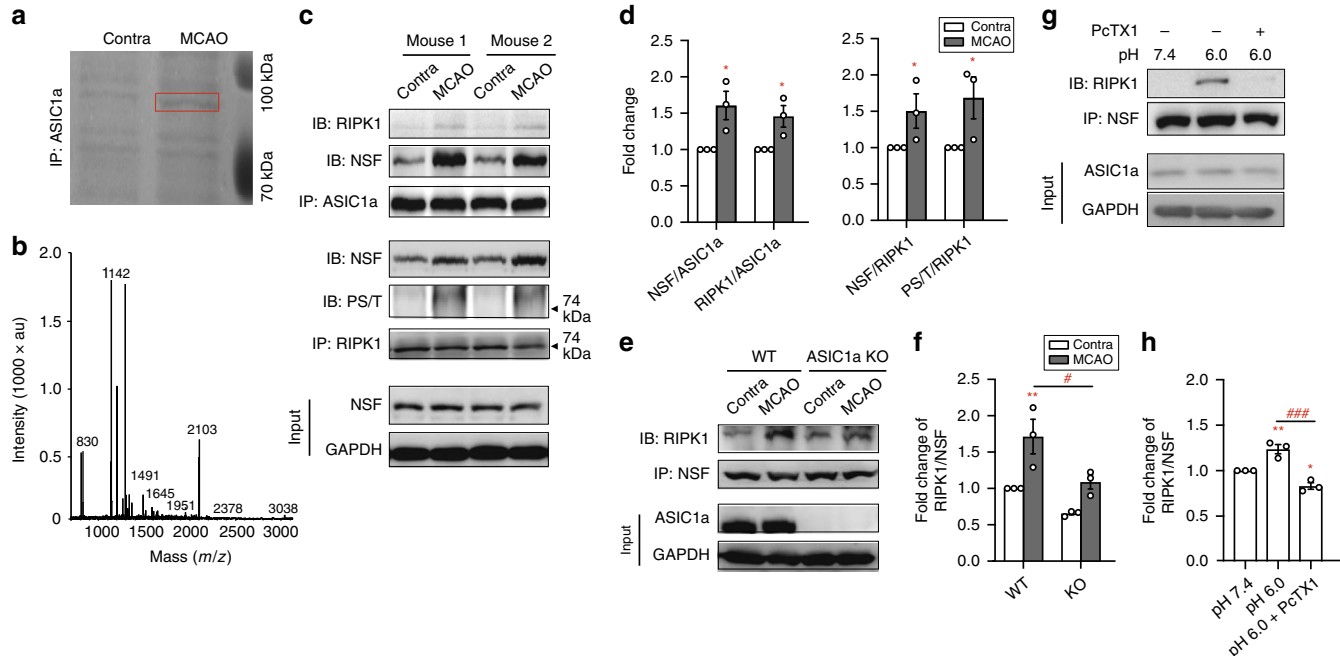

**Fig. 3 NSF is recruited to ASIC1a in ischemic brain to form NSF-ASIC1a-RIPK1 complex. a** Image of Coomassie blue staining of SDS-PAGE of immunoprecipitants by a polyclonal ASIC1a antibody from contralateral (Contra) and ischemic (MCAO) hemispheres of a mouse subject to MCAO. Red box indicates the band of interest (~85 kD) reproducibly identified in three independent experiments. **b** NSF was identified from the ~85 kD band by MALDI-TOF-MS analysis, in which 18 peptide sequences (out of 39) were found to match NSF (GenBank accession no. 18195) with the protein sequence coverage of 22% (Supplementary Data 1). **c** Confirmation of increased RIPK1-ASIC1a-NSF association in MCAO hemispheres as compared to Contra hemispheres by co-IP. In two examples, both the anti-ASIC1a and anti-RIPK1 antibodies pulled down more NSF from MCAO than Contra groups, along with the markedly increased serine/threonine phosphorylation of RIPK1. Total levels of NSF were not changed after MCAO. **d** Summary data for (**c**). $n = 3$, * $p < 0.05$ vs. Contra by ANOVA. **e** Asic1a gene deletion prevented the ischemia-induced increase in RIPK1-NSF association. Anti-NSF antibody pulled down more RIPK1 from MCAO than Contra brain samples from WT but not ASIC1a knockout (KO) mice. **f** Summary data for (**e**). $n = 3$, ** $p < 0.01$ vs. Contra, # $p < 0.05$ vs. WT, by ANOVA. **g** Acidosis (pH 6.0, 1 h) induced RIPK1-NSF association in cultured neurons, which was prevented by pretreatment with PcTX1 (10 nM added at 0.5 h before and maintained during acidosis). **h** Summary data for (**g**). $n = 3$, * $p < 0.05$, ** $p < 0.01$, vs. pH 7.4, ### $p < 0.05$ vs. pH 6.0, by ANOVA. Error bars are SEM for all summary panels.

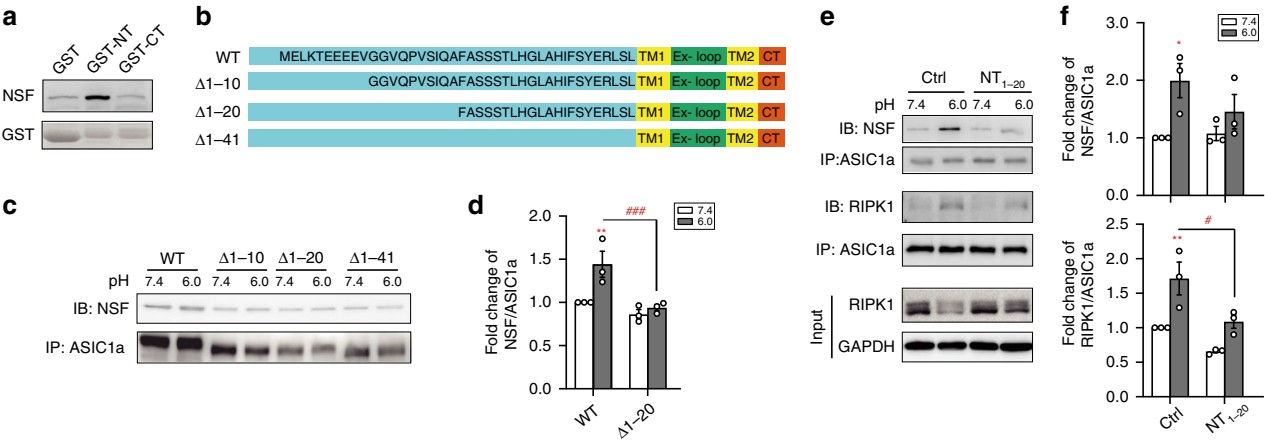

**Fig. 4 NSF binds to the distal N-terminus of ASIC1a. a** Glutathione-S-transferase (GST) pull-down assay showing the pulldown of NSF by GST-NT, but not GST-CT. Purified GST, GST-ASIC1a C-terminus (GST-CT) or GST-ASIC1a N-terminus (GST-NT) were incubated with mouse brain extracts, immobilized on glutathione-Sepharose 4B beads, and immunoblotted for NSF (top) and GST (bottom). GST alone served as a negative control. **b** Amino acid sequences of NT deletion mutants of ASIC1a. **c** Co-IP experiments showing that deletion of aa1-41 (Δ1-41) or aa1-20 (Δ1-20) decreased the acid-induced association between ASIC1a and NSF in CHO cells. **d** Summary data of (**c**). $n = 3$, ** $p < 0.01$ vs. corresponding pH 7.4, ### $p < 0.001$ vs. WT, by ANOVA. **e** Co-IP experiments showing that NT$_{1-20}$ (10 μM, 0.5 h pretreatment) reduced acid-induced formation of NSF-ASIC1a-RIPK1 complex in cultured mouse cortical neurons. NT$_{1-20}$ also decreased acid-induced degradation of RIPK1 protein (lower panel). **f** Summary data of (**e**). $n = 3$, * $p < 0.05$, ** $p < 0.01$ vs. corresponding pH 7.4, # $p < 0.05$ vs. WT, by ANOVA. Error bars are SEM for all summary panels.

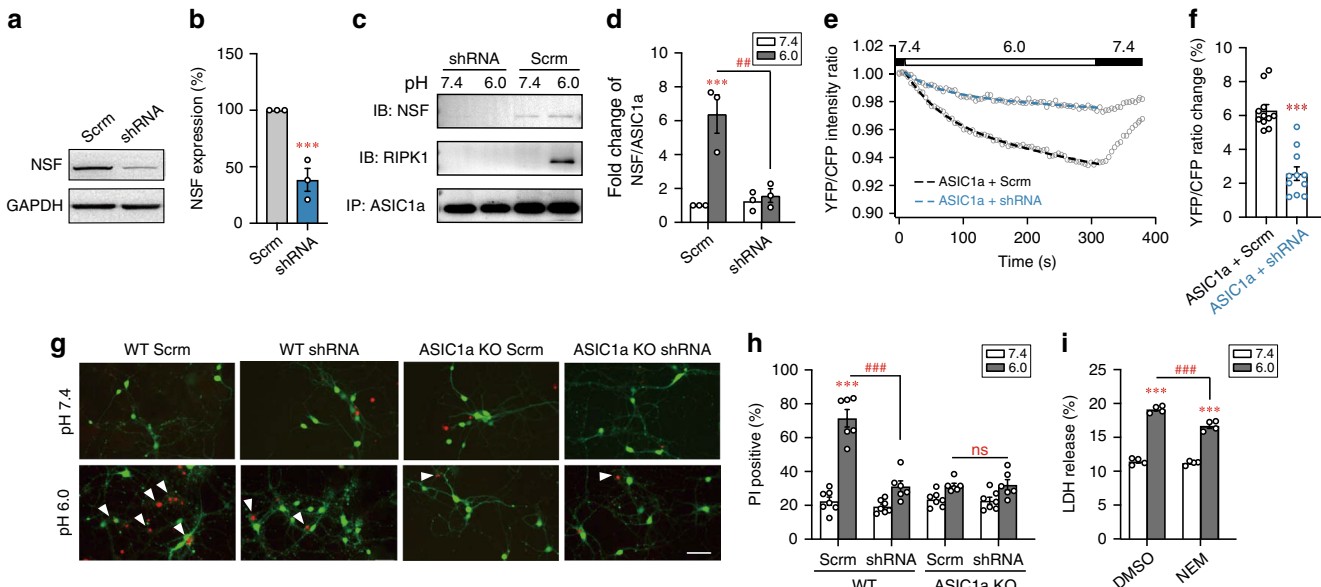

**Fig. 5 NSF knockdown protects against acidotoxicity in cultured neurons. a**, **b** Knockdown efficiency of NSF shRNA in cultured neurons determined by western blotting. Shown are representative blots (**a**) and summary data (**b**). $n = 3$, *** $p < 0.001$ vs. Scrm (scrambled shRNA) by paired $t$ test. **c** shRNA knockdown of NSF attenuated acid-induced associations of ASIC1a with NSF and ASIC1a with RIPK1. **d** Summary data of NSF pulled down by the ASIC1a antibody. The data are normalized to NSF/ASIC1a of neurons treated with Scrm at pH 7.4. $n = 3$, *** $p < 0.05$ vs. corresponding pH 7.4, ## $p < 0.01$ vs. Scrm in pH 6.0, by ANOVA. **e** NSF knockdown attenuated acid-induced decrease of CFP-ASIC1a-YFP FRET. Similar to Fig. 2b, but the cells were transfected with Scrm or NSF shRNA as indicated. $n = 11$ cells for each. **f** Summary of ratio changes at 5 min of the pH 6.0 treatment for data in (**e**). *** $p < 0.001$ vs. CFP-ASIC1a-YFP with Scrm by unpaired $t$ test. **g** PI staining of cultured cortical neurons prepared from WT and ASIC1a KO mice transfected with NSF-shRNA or Scrm shRNA. Scale bar, 50 μm. The neurons were treated at either pH 7.4 or pH 6.0 for 1 h and then returned to the normal culture medium for 24 h before PI staining. Knocking down NSF reduced acid-induced neuronal death in WT, but not ASIC1a KO, neurons. **h** Summary data for (**g**). $n > 200$ neurons counted for each. *** $p < 0.001$ vs. corresponding pH 7.4, ns, no statistical significance, ### $p < 0.001$ vs. Scrm in pH 6.0, by ANOVA. **i** The nonspecific NSF inhibitor, NEM (3 μM), showed a protective effect on acid-induced neuronal death. $n = 4$, *** $p < 0.001$ vs. corresponding pH 7.4, ### $p < 0.001$ vs. vehicle (DMSO) pH 6.0, by ANOVA. Error bars are SEM for all summary panels. The Scrm sequence used was CTTAAGGTTAAGTCACTCT.

settings) at pH 7.4 as compared to WT (Fig. 6b). The expression of E/A mutant in CHO cells also resulted in increased cell death, increased ASIC1a-NSF and ASIC1a-RIPK1 associations, and decreased expression level and proton-evoked whole-cell currents of ASIC1a, as compared to WT (Fig. 6c–g and Supplementary Fig. 5). These results strongly suggest that the negatively charged E$^6$EEE$^9$ at ASIC1a-NT are crucial for interactions with the CT death domain, creating auto-inhibition under resting conditions.

As shown earlier, peptide NT$_{1–20}$ protected cultured cortical neurons from acidosis-induced death to a similar degree as the RIPK1 inhibitor, Nec-1 (20 μM). By contrast, the peptide bearing the E → A substitutions (NT$_{1–20}$$^{E/A}$) failed to exert any neuroprotection (Fig. 6h). Given that NT$_{1–20}$ effectively attenuated acidosis-induced neuronal death in vitro, we asked if the peptide is also neuroprotective in vivo under conditions when acidotoxicity, especially ASIC1a, is known to play a role. Thus, we applied peptide NT$_{1–20}$ bilaterally to mouse lateral ventricles through microinfusion at 30 min before MCAO. Compared to vehicle-infused controls, peptide NT$_{1–20}$ significantly reduced ischemia-induced brain damage, as assessed by infarct volumes (Fig. 6i, j). The neuroprotective effect of peptide NT$_{1–20}$ was similar to that achieved by *ASIC1a* gene deletion; however, peptide NT$_{1–20}$$^{E/A}$ failed to show any protection against the ischemic damage (Fig. 6i, j). These in vivo experiments highlight again the importance of N- to C-terminal interaction of ASIC1a in suppressing acidosis-induced neuronal damage and critical involvement of negatively charged glutamates at ASIC1a distal NT for this interaction. They also demonstrate that targeting such interaction can be of potential therapeutic value in neurological disorders where acidotoxicity is a major contributor of neuronal damage.

## Discussion

The present study extends our earlier finding that acidosis-induced neuronal death involves an ion conduction-independent conformational coupling between ASIC1a and RIPK1 and consequent activation of necroptotic death[2]. We show here that such coupling is effectively shut-off under normal physiological conditions because of auto-inhibition exerted by ASIC1a-NT, which, at resting or closed state, masks the CT death motif through electrostatic interactions between NT glutamate and CT lysine residues. Interestingly, however, even though acid stimulation can disrupt the N- to C-terminal interaction to allow the CT death motif to recruit RIPK1, acidosis alone is insufficient to induce cell death unless NSF is present. NSF binds to ASIC1a distal NT in a state-dependent fashion, exhibiting increased association under acidosis and brain ischemia (Fig. 7). Plausibly, by binding to ASIC1a NT, NSF helps keep the CT death motif free to interact with RIPK1, allowing continued RIPK1 activation. More importantly, by maintaining NT-CT interactions with a synthetic peptide representing the first 20 amino acids of ASIC1a, we achieved neuroprotection in both in vitro model of acidotoxicity (Figs. 1d–h and 6h) and in vivo model of ischemic brain injury (Fig. 6i, j).

Traditionally, the ion conducting function of ASIC1a, especially Ca$^{2+}$ influx, which can cause Ca$^{2+}$ overload, was thought to be the main reason for neuronal acidotoxicity. Recently, we presented evidence that ASIC1a recruits RIPK1 to trigger RIPK1 self-phosphorylation and activation through a death motif located in ASIC1a proximal CT, independently of the channel's ion-conducting function[2]. Although conductance-independent functions of ion channels have gained greater appreciations these days[10–12,22–25], mechanistic details that differentiate the ion

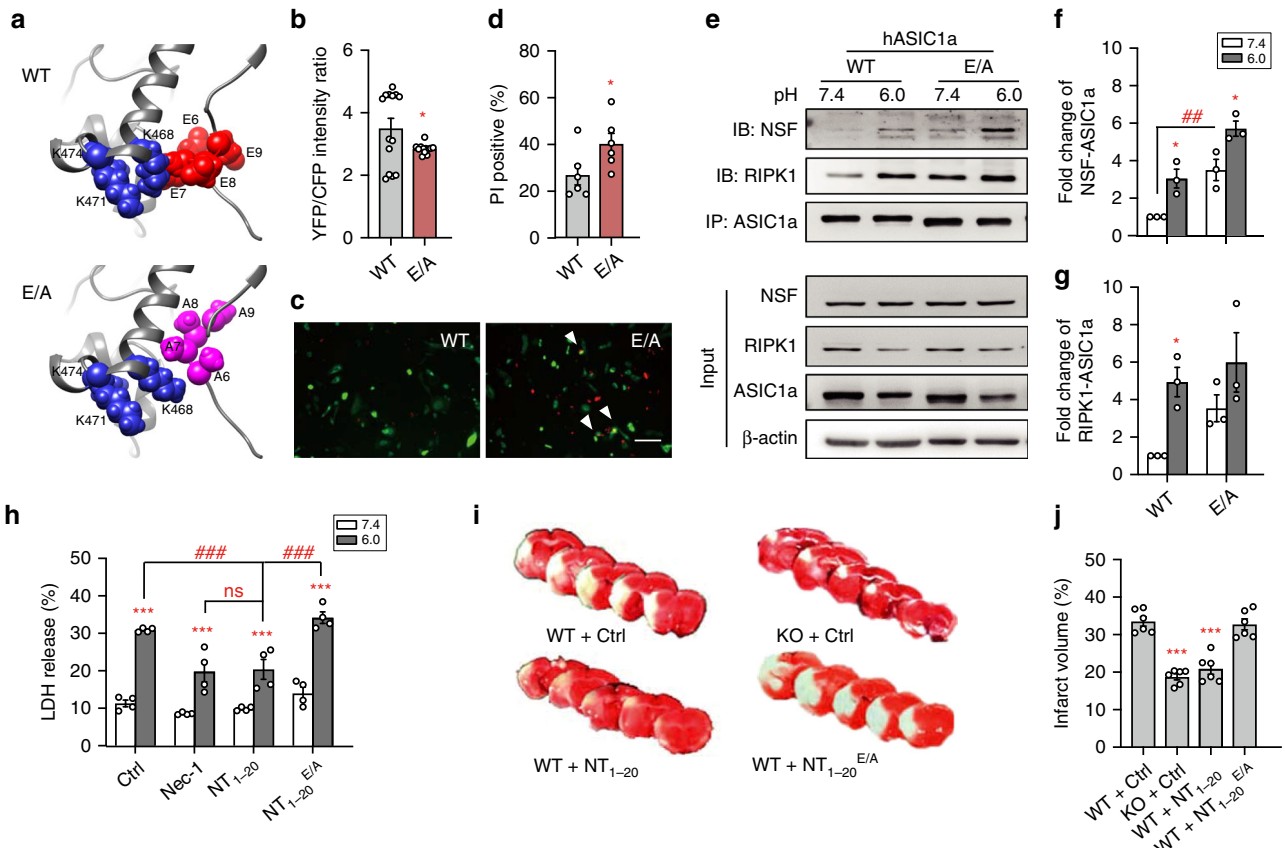

**Fig. 6 E⁶EEE⁹ at distal N-terminus of ASIC1a are critical for intracellular auto-inhibition. a** Rosetta model of closed state ASIC1a suggests that negatively charged glutamate residues (E$^6$EEE$^9$) are juxtaposed to positively charged lysines (K$^{468}$, K$^{471}$, and K$^{474}$) in the CT death motif, forming electrostatic interactions (upper). The distance between beta carbons of E$^7$ and K$^{468}$ is ~4.7 Å. With E → A substitutions at E$^6$EEE$^9$ (lower), ASIC1a-NT moves apart from the lysines at CT death motif. (**b**) The E/A mutant exhibited lower FRET than WT ASIC1a at pH 7.4. CFP and YFP were tagged at NT and CT, respectively. The YFP/CFP emission ratios were acquired under the same conditions using identical settings. $n = 12$–14, * $p < 0.05$ vs. WT by unpaired $t$ test. **c, d** Expression of E/A mutant in CHO cells resulted in more cell death than WT ASIC1a at pH 7.4. Representative images of PI-stained cells (**c**) and quantification of PI-positive CHO cells (**d**). $n > 200$ transfected neurons (green) counted for each. Scale bar, 20 μm. * $p < 0.05$ vs. WT by unpaired $t$ test. **e** Co-IP experiments of transfected CHO cells showing that E/A mutant exhibited increased association with NSF and RIPK1 at pH 7.4. **f, g** Summary data for ASIC1a-NSF (**f**) and ASIC1a-RIPK1 (**g**) association as determined in (**e**). Data are normalized to NSF/ASIC1a and RIPK1/ASIC1a of WT group at pH 7.4, respectively. $n = 3$, * $p < 0.05$ vs. corresponding pH 7.4, ## $p < 0.01$ vs. WT, by ANOVA. **h** LDH assay showing that the E → A substituted NT$_{1-20}$ peptide (NT$_{1-20}$$^{E/A}$, 10 μM) failed to exert neuroprotection against acidotoxicity of cultured cortical neurons, contrasting to Nec-1 (20 μM) and peptide NT$_{1-20}$ (10 μM). $n = 4$, *** $p < 0.001$ vs. corresponding pH 7.4, ### $p < 0.001$, ns, no statistical significance vs. NT$_{1-20}$ at pH 6.0, by ANOVA. **i, j** Representative images of brain sections (**i**) and summary data (**j**) showing that pre-administration of peptide NT$_{1-20}$, but not peptide NT$_{1-20}$$^{E/A}$, by microinfusion reduced infarct volumes in mouse MCAO model. KO is ASIC1a KO. $n = 6$, *** $p < 0.001$ vs. WT with Ctrl peptide, by ANOVA.

transport function and conformational signaling of an ion channel have only been illustrated in few cases, including for NMDA receptors, an agonist-driven movement of the cytoplasmic domain[26] and tyrosine dephosphorylation[27], and for Ca$_V$1.2, the voltage-dependent conformational change vital to CaMKII activation[11]. Often, the same cue (ligand or voltage) for channel gating is also the trigger of conformational signaling.

Conformational signaling may represent a simple and ancient mechanism for signal transduction[28]. Here, we reveal auto-inhibition through the cytoplasmic NT-CT interaction that keeps the CT death motif of ASIC1a from eliciting cytotoxicity at rest. At low pH, the interaction is disrupted to free the death motif, which in turn activates RIPK1. Proton-induced conformational changes of ASIC1a have been well-documented by high-resolution X-ray crystallographic and cryo-electron microscopic structures that represent open, desensitized, and resting states of the channel[13,14,17]. However, all these structures are truncated, lacking 13–25 NT and up to 64 CT residues[17]. We found that deletion of just the first 20 residues from ASIC1a leads to a huge

reduction of proton-induced currents (Fig. 2g) and deletion of the first 10, 20, or all NT residues, or just neutralization of the four glutamate residues there, also results in reduced protein expression (Figs. 2h, 4c; Supplementary Figs. 1a, b and 4). These suggest that either the NT facilitates folding and/or maintains protein stability of ASIC1a or there is an activity-dependent degradation of this protein. Supporting the idea that ion-conductance and conformational signaling of ASIC1a are differentially regulated, the non-conducting mutant HIF mediates cell death just like WT ASIC1a in response to either acid treatment[2] or the first 20 NT residue deletion (Fig. 2e).

Acid triggers ASIC1a channel gating and conformational signaling with very different kinetics. Whereas the current develops very quickly and then inactivates almost completely within 5 s, the FRET decrease, reflective of NT-CT dissociation, takes at least 5 min to reach a quasi steady-state and exhibits no obvious inactivation. A mismatch in time courses of ASIC1a-mediated ionic currents and neuronal death had also been noticed before, as the degree of neuronal cell death depends on the duration of

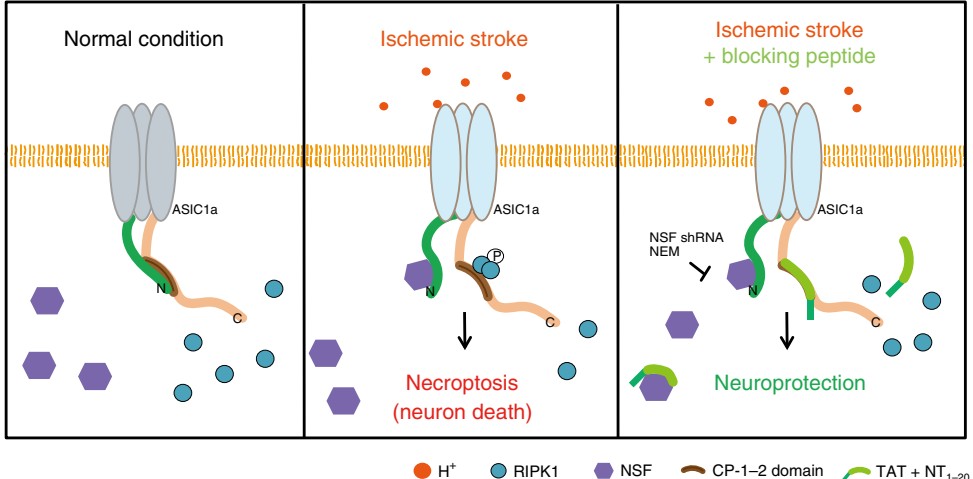

**Fig. 7 Schematic diagram of proposed mechanism underlying conformational signaling of ASIC1a in acidotoxicity.** Under normal conditions, the CP-1 death motif of ASIC1a-CT is masked by interaction with ASIC1a-NT (left). Acidosis activates ASIC1a and causes conformational transition that disrupts the preformed N- to C-terminal interaction within ASIC1a to unmask the CT death motif. This transition is facilitated and stabilized by NSF binding to ASIC1a-NT. The freed ASIC1a-CT recruits RIPK1 and activates it by phosphorylation, resulting in neuronal necroptosis (middle). Peptide NT$_{1-20}$ binds to ASIC1a-CT to prevent recruitment and thereby activation of RIPK1 to achieve neuroprotection (right). NT$_{1-20}$ may also bind to NSF, decreasing its availability for ASIC1a-NT, similar to NSF shRNA knockdown or inhibition by NEM.

acidosis treatment in minutes or even hours[2]. Our FRET results, showing lack of reassociation between ASIC1a-NT and CT during continued acidosis, is more consistent with the time dependence in severity of acidotoxicity. Therefore, the proton-evoked persistent NT-CT dissociation revealed by FRET likely reflects the actual conformational change underlying ASIC1a-mediated acidotoxicity. Admittedly, the large sizes of the reporter fluorophores used here could pose interference on the NT-CT interaction as both termini are very short. Future studies using smaller fluorophores such as transition metal ion FRET with unnatural amino acid[29] should help ameliorate this issue. Based on the previous work and our new data, the long-term dissociation of NT probably provides the exposed CT death motif with an extended ability to recruit more RIPK1 over time, yielding the observed time dependence on acidosis-induced death.

More interestingly, acid-induced conformational change of ASIC1a represents only the initial event. The NT-CT dissociation may be intrinsically transient and prone to quick reversal in the absence of a molecular chaperone. Here, we found NSF to serve this function. NSF has traditionally been described as a chaperone that facilitates disassembly and recycling of soluble NSF attachment protein–receptor (SNARE) proteins[30]. Recent studies suggest that NSF also serves as a structural chaperone of other proteins such as receptors and protein kinases, including AMPA receptors, β2 adrenergic receptors and GABA$_{A1}$ receptors, by modulating their trafficking or recycling[31–33]. The present study reveals another partner, ASIC1a, to which NSF binding at distal NT helps stabilize the dissociation of ASIC1a-NT from CT and thereby sustain conformational signaling over time. In addition to offering molecular details on acidosis-induced and ASIC1a-mediated activation of RIPK1 to trigger necroptosis, these findings also provide insights into the diverse function of NSF, with far-reaching implications in designing therapeutic strategies against a broad spectrum of neurological diseases.

Auto-inhibition is a widespread and often constitutive mechanism that represses spontaneous activation of a potentially damaging signal[34]. Intramolecular auto-inhibition has been well-described for many receptors and kinases, such as 5-HT$_{1A}$ receptor, EphB2 receptor, CaMKII, and Ca$^{2+}$-ATPase[35–39], as well as ion channels[40]. Auto-inhibition often involves the

inhibitory domain juxtaposing and sterically masking the active domain, and unmasking is achieved by a conformational change during activation[34]. ASIC1a also adheres to this rule. Although three-dimensional structure is unavailable for ASIC1a NT and CT, our computational model and FRET examination support the idea that auto-inhibition of ASIC1a cytotoxicity involves a close interaction between distal NT and proximal CT death motif of the channel and their separation in response to acidosis. This view is further supported by the finding that NT deletions caused cell death without acidosis, demonstrating that a main acid-induced conformational effect is to remove auto-inhibition of ASIC1a cytotoxicity. In the Rosetta model, ASIC1a-NT is predicted to interact with the CT death motif, yielding a distance of ~4.7 Å between beta carbons of E$^7$ and K$^{468}$ in the basal state; FRET results also suggest a distance of <10 nm between N- and C-terminal ends. Furthermore, disrupting the charged residues, E$^6$EEE$^9$, in these regions by neutralization led to cell death.

The decipherment of auto-inhibition mechanism of ASIC1a provides an opportunity of neuroprotection using an inhibitory peptide that specifically targets acidosis-induced neuronal damage without affecting channel's ionic conduction (Fig. 1i). Peptide-based neuroprotectants have become attractive for their specificity, efficiency and low cost, and some candidate peptides have advanced to phase III clinical trials[41,42]. Here, we show the neuroprotective effect of peptide NT$_{1-20}$ in several conditions. First, it antagonized the detrimental action of peptide CP-1-2 (Fig. 1c), which, like previously shown for CP-1, induces death through activation of RIPK1[2]. This finding provides a strong argument that the binding between the two cytoplasmic regions effectively prevents the recruitment of RIPK1 by the ASIC1a-CT death motif. Second, NT$_{1-20}$ attenuated acidosis-induced death of cultured cortical neurons (Figs. 1d–h, 6h). This was accompanied with detectable decreases in acid-induced association between ASIC1a and RIPK1, degradation of RIPK1, and association between ASIC1a and NSF (Fig. 4e, f), suggesting again that the peptide exerts its effect through binding to the CT death motif and thereby preventing RIPK1 activation. Here, the peptide might have an added effect on sequestering NSF, which could also impair ASIC1a-RIPK1 association during acidosis. Both actions, i.e., masking the death motif at ASIC1a-CT and sequestering NSF,

would result in neuroprotection against acidosis. Finally, peptide $NT_{1-20}$, but not its inactive substituted version, $NT_{1-20}^{E/A}$, protected brains from ischemic injury to a similar extent as ASIC1a gene ablation in mice (Fig. 6i, j). This in vivo finding further argues for the neuroprotective potential of $NT_{1-20}$ in ischemic brain damage. As ASIC1a plays important roles in synaptic plasticity, learning and memory, targeting the auto-inhibition of ASIC1a without impairing its physiological function should be desirable in stroke therapies.

In conclusion, we elucidate a molecular mechanism involved in conformational signaling of ASIC1a that causes neuronal necroptosis in response to tissue acidosis, a prevalent effect associated with ischemic stroke and many other neurological disorders. By demonstrating an auto-inhibition mechanism that involves an interaction between ASIC1a-NT and CT, we uncovered the critical role of negatively charged glutamate residues located at distal NT of ASIC1a in keeping the channel quiescent and the crucial involvement of NSF in preventing auto-inhibition. We further demonstrate the possibility and therapeutic potential of targeting the ASIC1a-RIPK1 pathway for neuroprotection against acidotoxicity making use of the auto-inhibitory mechanism. Our study sheds lights on new and specific ways to target acidotoxicity for therapeutic intervention of ASIC1a-related illnesses, which include multiple types of neurodegenerative diseases.

## Methods

**Animals**. Animal care and the experimental protocols were approved by the Animal Ethics Committee of Shanghai Jiao Tong University School of Medicine, Shanghai, China (permit number: DLAS-MP-ANIM.01-05). The WT C57BL/6J mice were obtained from Shanghai Slac Laboratory Animal Company, China. The conventional global ASIC1a KO mice (with a congenic C57BL/6 background) were the generous gift of Professor Michael J. Welsh (Howard Hughes Medical Institute, University of Iowa, Iowa City, IA, USA). To reduce the experimental variability, age-matched littermate pairs resulting from heterozygous crossings were used for all experiments involving ASIC1a KO mice.

**Focal ischemia**. The experimental protocols (ethics protocol number: 2015011) were approved by the Animal Care and Use Committee of Shanghai Jiao Tong University School of Medicine, Shanghai, China. Briefly, animals (male C57BL/6 mice, ~25 g) were anesthetized using 5% chloral hydrate in normal saline. Rectal temperature was maintained at $37 \pm 0.5\,°C$ with a thermostatically controlled heating pad. The monofilament (702234PK5Re, Docclo Ltd.) was advanced to block the origin of the MCA, while cerebral blood flow was stably reduced to below 20%, as monitored by transcranial laser Doppler (Moor Instruments Ltd.)[2]. For co-IP experiments, after decapitation, brains were removed immediately after 1 h of MCAO. In TTC (2, 3, 5-triphenyltetrazolium hydrochloride) staining for detection of experimental cerebral infarction, the mice were euthanized after 24 h of reperfusion, and brains were sectioned coronally at 1 mm interval after short freezing and stained with the vital dye TTC (in normal saline) incubated in the incubator at 37 °C for 5 min at each side in the dark and then placed in 4% paraformaldehyde for fixation. Viable tissues are stained red while dead tissues are left unstained. After fixation for 24 h, images were taken for all brain sections and the percentage of unstained portions (infarct volume) of each whole brain was determined by analysis using ImageJ.

**Surgical procedures and drug microinjection**. Animals anaesthetized with 5% chloral hydrate in normal saline and placed in a stereotaxic apparatus (RWD Life Science) were implanted bilaterally with a guide cannula with stylus aimed to the paired lateral ventricles following the coordinates below: anteroposterior, −0.5 mm; lateral, ±1.0 mm; dorsoventral, −2.5 mm. The cannulas were positioned in place with acrylic dental cement and secured with skull screws. A stylus was placed in the guide cannula to prevent clogging[43]. Animals were allowed to recover from surgery for a week before experimental manipulations. For microinfusion, the stylus was removed from the guide cannula, and an infusion cannula connected to a microsyringe via a tubing was inserted. Peptides (50 mM in artificial cerebrospinal fluid, 3 μl per side) were microinfused into lateral ventricles through the microsyringe driven by a microinfusion pump (KDS 310, KD Scientific) at the rate of 0.2 μl/min.

**Molecular biology and cell transfection**. The complementary DNA (cDNA) of mouse ASIC1a (GenBank accession: NM_009597.1) in pEGFP-C3 vector (Clontech) was expressed in CHO cells (GNHa 3, Shanghai Institute of Biochemistry and

Cell Biology, Chinese Academy of Sciences) by transient transfection as reported previously[44]. Deletion mutations were made by PCR with primers matching the sequences excluding the respective deletion target and confirmed by DNA sequencing. Primers used in the plasmid construction include, for Δ1–10, TATCTCGAGATGGGTGGCGTCCAGCCGGTGA and TATAAGCTTTCAGCA GGTAAAGTCCTCGAAC; for Δ1–20, TATCTCGAGATGTTCGCCAGCAGCT CCACAC and TATAAGCTTTCAGCA GGTAAAGTCCTCGAAC; for Δ1–41, TATCTCGAGATGAAGCGGGCACTGTGGGCCC and TATAAGCTTTCAGC AGGTAAAGTCCTCGAAC. CHO cells were cultured in 0# glass with F12 medium supplemented with 2 mM L-glutamine and 10% fetal bovine serum (FBS, Gibco), and transiently transfected with cDNA constructs using Hilymax (Dojindo laboratories) following the manufacturer's protocol. Cells were used within 1–2 days after transfection.

**Primary culture of mouse cortical neurons**. Postnatal day 1 C57BL/6 WT or ASIC1a KO mice were anesthetized with halothane. Brains were removed rapidly and placed in ice-cold $Ca^{2+}$- and $Mg^{2+}$-free phosphate buffered saline (PBS). Tissues were dissected and incubated with 0.05% trypsin-EDTA for 15 min at 37 °C, followed by trituration with fire-polished glass pipettes, and plated in poly-D-lysine-coated culture dishes or 24-well plates. Neurons were cultured with Neurobasal medium supplemented with B27 and maintained at 37 °C in a humidified 5% $CO_2$ atmosphere incubator[2]. Cultures were fed twice a week and used in 14–16 days after plating.

**Co-immunoprecipitation**. Transfected CHO cells ($2 \times 10^6$ cells), cultured mouse cortical neurons (~$10^7$ cells) or brain tissues (200–250 mg) were collected and resuspended in a lysis buffer containing 20 mM Tris-Cl, pH 7.4, 150 mM NaCl, 1% Triton X-100, 1 mM EDTA, 3 mM NaF, 1 mM β-glycerophosphate, 1 mM sodium orthovanadate, 10% glycerol, complete protease inhibitor set (Sigma-Aldrich), phosphatase inhibitor set (Sigma-Aldrich) and 2 mM N-ethylmaleimide. The resuspended lysates were vortexed, incubated on ice for 40 min, and centrifuged at $13,000 \times g$ for 15 min. The supernatant was incubated with 4 μg antibody at 4 °C overnight. Protein G agarose beads (20 μl, Thermo Fisher Scientific) were added to the sample and incubated for 2 h at 4 °C. Then, immunoprecipitants were washed three times with the lysis buffer, suspended with 2 × loading buffer, boiled, and run on SDS-PAGE, followed by transfer to PVDF membrane for immunoblotting as described[2]. Aliquots of the original lysates were also run on SDS-PAGE in parallel for immunoblotting to determine the amount of input. The primary antibodies used are anti-ASIC1a (1:500, sc-13905, Santa Cruz), anti-NSF (1:1000, ab18903, Abcam), anti-RIPK1 (1:1000, 610459, BD), anti-phosphor-S/T (1:500, #9624, Cell Signaling Technology), β-actin (1:1000, MAB1501, EMD Millipore), and anti-GAPDH (1:3000, KC-5G4, KangChen). This was followed by horse radish peroxidase-conjugated secondary antibodies (1:5000, Goat anti-rabbit IgG, AP132H; 1:5000, Rabbit anti-mouse IgG, AP160P; 1:5000, Rabbit anti-goat IgG, AP106P, EMD Millipore) and visualization with enhanced chemiluminescence on ImageQuant LAS 4000 mini digital imaging system (GE Healthcare Life Sciences). The dilution for all primary and secondary antibodies was in PBST with 1% BSA. The band intensities of western blots were quantified by ImageJ with background subtraction.

**GST pull down**. The cytoplasmic domains of human ASIC1a were produced as GST-fusion proteins in Escherichia coli BL21 (DE3) pLysS (TIANGEN). GST-fusion proteins were purified under nondenaturing conditions using GST Bind kits (Novagen) in accordance with the manufacturer's instructions. Extracts from mouse cerebral cortices were incubated with the purified GST-fusion proteins or GST alone immobilized on glutathione-Sepharose 4B (GE Healthcare) beads at 4 °C overnight in the co-immunoprecipitation lysis buffer. The next day, the beads were thoroughly washed and then resuspended with 2× loading buffer, boiled, and used for immunoblotting as described above.

**Electrophysiological recordings**. ASIC1a currents were recorded using conventional whole-cell patch-clamp techniques at room temperature (22–25 °C). The membrane voltage was held at −60 mV. The pipette solution contained: 120 mM KCl, 30 mM NaCl, 1 mM $MgCl_2$, 0.5 mM $CaCl_2$, 5 mM EGTA, 4 mM Mg-ATP, and 10 mM HEPES, pH 7.4, osmolarity kept at 280–300 mOsm/l. The standard external solution (SS) contained: 150 mM NaCl, 5 mM KCl, 1 mM $MgCl_2$, 2 mM $CaCl_2$, 10 mM glucose and 10 mM HEPES buffered to various pH values with Tris-base or HCl. The osmolarity of SS was kept at 300–330 mOsm/l. All drugs for electrophysiological experiments were purchased from Sigma–Aldrich.

**Forster resonance energy transfer**. The combination of CFP and YFP is the most popular FRET pair for studying protein-protein interaction and was used in this case. The FRET constructs (referred to as CFP-ASIC1a-YFP, or CFP-YFP for negative control) were transfected into CHO cells. The excitation wavelength for CFP donor is 405 nm. For Figs. 2b, 5e, 6b, FRET was monitored as the ratio between emission intensities of yellow fluorescence (525 nm) and cyan fluorescence (482 nm, $F_{YFP}/F_{CFP}$) in confocal laser scanning microscopy (A1, Nikon).

Because of the known issues with the use of crude $F_{YFP}/F_{CFP}$ ratio to assess FRET in live cells, such as the cross-talk, bleed-through, variable expression levels

of the fluorescence proteins among individual cells, and certain non-specific FRET signals[45], we also used spectra FRET (Fig. 2d), which allows for correction of these issues[45,46]. Spectra FRET was performed with an inverted fluorescence microscope (TE2000-U; Nikon) using a ×40 oil-immersion objective (NA 1.3). An argon laser (Spectra-Physics) was used to provide the excitation light, with the exposure time controlled by a shutter (Uniblitz; Vincent Associates) synchronized with the camera by software through an amplifier (PatchMaster; HEKA). Two filter cubes (Chroma Technology Corp.) containing the following elements were used (excitation filter, dichroic mirror, and emission filter): Cube I, Z488/20, Z488rdc, and HQ500LP; Cube II, Z514/10,Z514rdc, and HQ530LP. Spectra FRET began with the input slit of the spectrograph moved out of the light path, and the grating set at a small angle at which it is equivalent to a mirror projecting the cell image to the camera. The slit was then moved into the light path to cover the region of the cell from which fluorescence signals were to be measured. The grating was rotated to the desired angle for light of a selected wavelength range to be projected to the camera. To measure FRET from cells transfected with CFP-ASIC1a-YFP, we took two spectroscopic images, one with the CFP excitation at 436 nm and another with the YFP excitation at 500 nm. The fluorescence emission from 460 nm to 580 nm was recorded (Supplementary Fig. 2c). A fluorescence emission spectrum was then constructed from each spectroscopic image using the fluorescence intensity values along a horizontal line whose position corresponds to the part of the cell to be measured[47]. FRET efficiency from individual cells was calculated from the corrected YFP emission measured between 518 nm to 534 nm with the CFP excitation (436 nm)[45]. To correct for cross-talk and bleed-through signals of YFP and CFP, respectively, at neutral and acidic pH, we made separate calibrations for CHO cells expressing YFP or CFP when the cells were bathed in pH 7.4 and pH 6.0 solutions. For each pH, the same calibration values were used for CFP-ASIC1a-YFP, CFP-ASIC1a-HIF-YFP, and CFP-YFP to determine the FRET efficiencies.

**Surface biotinylation assay**. Surface biotinylation was performed on primary cultured neurons and CHO cells following established protocols[48]. Cells were incubated in pH 7.4 or pH 6.0 solution for 1 h. After washing three times with an ice-cold PBS+/+ (PBS plus 1 mM $MgCl_2$ and 2.5 mM $CaCl_2$) solution, cells were incubated with 0.25 mg/ml NHS-biotin (21335, Thermo Scientific) in the PBS+/+ solution at 4 °C for 30 min, and then washed with the PBS+/+ solution containing 0.1 M glycine to quench the reaction. Ten percent volume of the lysate was saved for determination of total (T) proteins and mixed with 4× loading buffer. The remainder, representing surface (S) proteins, was isolated by NeutrAvidin Agarose Resin (TD264106, Thermo Scientific) and eluted as described above for co-immunoprecipitation.

**Death assay**. First, cells were washed three times with SS buffered to pH 7.4 or 6.0, and then incubated in $CO_2$ atmosphere incubator for 1 h. The cells were then cultured in normal culture medium for 24 h at 37 °C with 5% $CO_2$. Cell viability was assessed by cell titer blue (CTB) assay, propidium iodide (PI) staining and lactate dehydrogenase (LDH) production. For the CTB assay, neurons were cultured in 24-well plates. The amount of culture medium was adjusted to the same in each well (0.5 ml) and pH 6.0 treatment was performed in the absence and presence of different drugs. After returning to normal culture for 24 h, 0.1 ml CTB solution (Promega Corporation) was added to each well and the plate incubated for 2 h at 37 °C. The fluorescence intensities (excitation, 560 nm; emission, 590 nm), indicative of the amounts of viable cells, were measured using the Synergy HTX multi-mode microplate reader (BioTek Instruments). For PI staining, cells were stained with 10 μg/ml PI for 15 min at 37 °C and then washed three times with SS. Cell morphology was observed through differential interference contrast which helps distinguish neurons from glia, and PI staining was examined by fluorescence microscopy. The LDH levels in the culture media, indicative of cell death, were tested with the LDH assay kit (Promega Corporation) following the manufacturer's protocol and measured using the Synergy HTX multi-mode microplate reader (BioTek Instruments). All death assays were performed with more than four biological replicates.

**shRNA of mouse NSF**. NSF shRNA was designed to target mouse NSF with the sequence GCTTCAATGATAAGCTCTT, and its effectiveness was tested by immunoblotting. shRNA with a scrambled sequence was used as a negative control. The sequence was driven by human synapsin I (hsynapsin I) promoter to specifically limit the expression to neurons. The cultured mouse cortical neurons were infected with lentivirus for NSF shRNA or the scrambled (Scrm) negative control shRNA at 7 days after plating. Assays were performed 7 days after virus infection.

**Peptides synthesis**. All synthetic peptides were acetate peptides and synthesized by solid-phase peptide synthesis provided by GL Biochem Ltd (Shanghai, China). The peptides were assembled with HIV-1 TAT protein transduction domain (GRKKRRQRRRC) to help penetration of cell membrane except for CP-1-2S. The Ctrl peptide contained only TAT sequence, CP-1 is TAT linked with LCRRGKC QKEAKRNSADKGVA, CP-1-1 is TAT linked with LCRRGKCQKEAKRN, CP-1-2 is TAT linked with KCQKEAKRN, CP-1-3 is TAT linked with KEAKRN, and CP-1-2S is KCQKEAKRN without TAT. $NT_{1-20}$ is TAT linked with MELKTEEEEV

GGVQPVSIQA, $NT_{1-20}^{E/A}$ is TAT linked with MELKTAAAAVGGVQPVSIQA, $NT_{11-30}$ is TAT linked with GGVQPVSIQAFASSSTLHGL, $NT_{21-41}$ is TAT linked with FASSSTLHGLAHIFSYERLSL, $NT_{1-10}$ is TAT linked with MELKTEEEEV, and $NT_{11-20}$ is TAT linked with GGVQPVSIQA.

**Mass spectrometry**. Proteins bound to ASIC1a were co-immunoprecipitated with the anti-ASIC1a antibody from ischemic hemisphere and contralateral hemisphere. Bound proteins were separated by SDS-PAGE and visualized by Coomassie blue staining. Bands of interest were excised from the gel, trypsin-digested, and subject to matrix-assisted laser desorption/ionization time-of-flight mass spectrometry (MALDI-TOF-MS) analysis.

**Intracellular pH measurement**. CHO cells grown on glass coverslips were incubated with 5 μM BCECF-AM at 37 °C for 30 min and then washed twice with SS (pH 7.4). The coverslip was transferred to a perfusion chamber, which was mounted on the stage of a Nikon Eclipse TI microscope (Japan). BCECF fluorescence images were taken with alternating excitation wavelengths of 488 nm and an emission wavelength of 535 nm at 0.5 Hz while the bath solution was changed from pH 7.4 to pH 6.0 or 10 μM cariporide was applied in the pH 7.4 solution. F0 was defined as the fluorescence in pH 7.4.

**Statistical analysis**. During data collection, the experimenters were blind to group allocation to avoid experimenter bias. Data were processed and analyzed by SPSS 19.0 software (SPSS Inc., Chicago, IL, USA) and expressed as mean ± SEM. Student's $t$ test was used to compare the results between two groups and one-way ANOVA and two-way ANOVA were used to assess the difference between multiple groups. $p < 0.05$ was considered statistically significant and the exact p values are presented in Supplementary Data 2.

**Reporting summary**. Further information on research design is available in the Nature Research Reporting Summary linked to this article.

## Data availability

Data supporting the findings of this manuscript are available from the corresponding authors upon reasonable request. A reporting summary for this article is available as a Supplementary Information file. The source data underlying Figs. 1b, c, e, f, h, 2b–d, f–i, 3a, c–h, 4a, c–f, 5a–f, h, i, 6b, d–h, j and Supplementary Figs. 1c–e, 2d, e, 3a–c, 4a, 5a, b, d, e are provided as a Source Data file.

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

## Acknowledgements

We thank Wei-Guang Li for fruitful discussion. We thank Jun-Long Huang, Xiao-Kun Zuo, Zhen-Yu Cai and Qin Jiang for technical support. This study was supported by grants from the National Natural Science Foundation of China (81730095, 81671130, 31800853, and 31461143004), the Innovative Research Team of High-level Local Universities in Shanghai, the Science and Technology Commission of Shanghai Municipality (18JC1420302), the Shanghai Municipal Science and Technology Major Project (2018SHZDZX05) and the US National Institutes of Health (NS102452). Jing-Jing Wang was a postdoctoral fellow supported by funding from the China Postdoctoral Science Foundation (M610259).

## Author contributions

J.-J.W., Y.-Z. W., Q.H., M.X.Z. and T.-L.X. designed the project. J.-J.W., F.L. and X.Q. performed western blotting. F.L. and F.Y. performed Forster resonance energy transfer. J.-J.W. and F.L. performed cell death assay. J.-J.W. and F.L. did electrophysiological recordings and analysis. J.-J.W. and F.L. prepared the transient focal ischemia models and drug microinjection. X.Q. and Y.L. prepared primary culture of mouse cortical neurons. J.-J.W., Q.H. M.X.Z. and T.-L.X., wrote the manuscript. All authors read and approved the final manuscript.

## Competing interests

The authors declare no competing interests.
