## [Peer Review File · Nature Communications]

Reviewers' Comments:

Reviewer #1:

Remarks to the Author:

Wang et al. previously showed (E-Life, 2015) that a direct interaction of the C-terminus of acid-sensing ion channel (ASIC) 1a with receptor-interacting protein kinase (RIP) 1 leads to phosphorylation of RIP1 and ensuing necroptosis. In the present manuscript, they extend these findings showing that the interaction of the ASIC1-C-terminus with RIP1 is prevented at neutral pH by a tight interaction of the ASIC1 C-terminus with the ASIC1 N-terminus. This interaction is slowly released (with a time course of a few minutes) by extracellular acidification, allowing association of the C-terminus with RIP1. Moreover, they show that the N-ethylmaleimide sensitive factor (NSF) interacts with the free N-terminus of ASIC1a to prevent its re-association with the C-terminus. These protein interactions are all independent of ions permeating through the ASIC1 pore. The authors come up with a model in which they claim that conformational changes triggered by extracellular acidification lead to the slow dissociation of N- and C-termini of ASIC1, followed by their association with NSF and RIP1, respectively, which would finally induce necroptosis. Thus, they propose a model in which ASIC senses protons not just to open its ion pore but to trigger intracellular downstream events that are independent of its ion channel function. Although to my knowledge the original study published in E-Life has not yet been reproduced by others and although some mechanistic details remain unclear, this is a highly original study which opens up a completely new view on the role of ASIC1a in ischemic neuronal death. I am convinced that it will inspire a lot of follow-up studies. Therefore, this manuscript is of great interest to the field of ASICs and of basic mechanisms in ischemic neuronal death. It also has a clear translational relevance.

I have a few comments to further improve this manuscript:

Major comments:

1)

The authors claim that conformational changes in ASIC1a triggered by extracellular acidification induce the slow dissociation of N- and C-terminus. The evidence for this hypothesis is rather indirect, however. An alternative explanation would be, for example, that the extracellular acidification leads to a slow intracellular acidification which then leads to the dissociation of N- and C-termini. Relatively easy ways to discriminate between these two possibilities would be, first, to follow the decrease in pHi by pH-imaging in CHO cells and in parallel to measure FRET like in Fig. 2b. Is the time course of both events similar? Second, the authors could try to reduce intracellular pH by other means (for example by blocking NHEs). Third, they could use mutants of ASIC1a that are no longer competent for conformational changes, for example due to engineered cysteine bonds. Such a mutant has been described by the Kellenberger group (E235C/Y389C; Gwiazda et al. JBC 2015). Is such a mutant still able to induce necroptosis, for example in CHO cells? Although these experiments will not provide a definite answer, they could help support the conclusion of an extracellular conformational change transmitted to the intracellular termini.

2)

The statistical analysis can be improved at several instances and more information on statistical analysis needs to be provided:

a)

Please mention unequivocally whether assays (for example the death assays) included technical replicates or only biological replicates and what a biological replicate exactly meant. On page 26 the authors state "All death assays were performed with more than four repeats each time" As some graphs show only four symbols, I guess what is really meant is "All death assays were performed with at least four repeats each time"?

b)

Was the experimenter really blind to group allocation for all assays as is mentioned in the

Reporting Summary? Please state this explicitly also in the Methods section under "Statistical analysis".

c)

It appears that in different graphs "control" means something different. In some graphs at least it is a control peptide. Which peptide has been used as a control? The authors need to mention its sequence and concentration. Please state in each case explicitly what the control was.

d)

Please explicitly state in figure legends (or in the text), which type of statistical analysis has been done (t test or ANOVA). Multiple t-tests using the same control need to be adjusted for multiple comparisons (for example, by Bonferroni correction).

e)

Experiments reported in figures 3c-e, 4c-d, and 5c (three repetitions) should be quantified and the quantitative summary data should be shown.

f)

Not always does the statistical analysis, which has actually been done, exactly support what is said in the text. For example, on page 13 it is said that "shRNA knockdown of NSF abolished acidosis-induced neuronal death, as shown by PI staining". But the statistics compares the difference in acidosis-induced cell death between two conditions. Acidosis actually seems to still induce cell death also in the shRNA group, although to a strongly reduced amount. (if this is the case, a statement like "shRNA knockdown of NSF attenuated acidosis-induced neuronal death" would be more appropriate). Another example: while on page 32 it is said that "CP-1-3 did not induce death", figure 1b shows significantly reduced viability in the presence of CP-1-3. Furthermore, still on page 32 it is said that "pH 6.0-induced cell death was significantly reduced by the pretreatment of NT1-20 but not control peptide." But the statistics compares NT1-20 vs control (and not NT1-20 and control vs. vehicle).

g)

The authors should consider providing exact P values (rather than, for example, $P < 0.01$).

h)

In figure 3a, a paired t test has been performed (pages 37/38). Since I doubt that identical neurons have been transfected with control shRNA and with NSF shRNA, the use of a paired t test is not appropriate in this case.

3)

In order to facilitate reproduction of the study, please mention the amount of neurons and the weight of brain tissue used for co-immunoprecipitation (Methods, page 23). Also, please clearly state how the standard medium (SS) was buffered to pH 6.0. Was the concentration of bicarbonate adjusted? Was another buffer (HEPES?) present (Methods, page 25)? Finally, please mention how many peptide sequences were identified by MALDI-TOF. (Page 34: "18 peptide sequences were found to match NSF" - 18 out of how many?) Any other hits that turned up several times?

Minor comments:

1)

Page 5: The numbering of the CP-1 peptide (amino acids 463-483) is correct but does not correspond to the numbering published in the E-Life paper. Perhaps the authors could clarify this discrepancy in a short note somewhere in the Methods.

2)

Page 6: any idea why the NT1-20 peptide lost its protective effect at a high concentration (20 microM)?

3)

Fig. 1c: any idea, why NT21-41 increased (rather than decreased) LDH release? If the increase

was significant, this should be indicated on figure 1c.

4)

Fig. 2a: the difference between the two conformations is difficult to see in the blow-up.

5)

Page 13 and page 36: Please change "positively changed lysine" to "positively charged lysine".

6)

Page 13: Does figure 6b really depict FRET efficiency? On page 8, it is explained that spectra FRET is needed to measure FRET efficiency.

7)

If I see it correctly, in the main text there is no reference to Supplementary figure 4.

8)

Page 14: "are crucial for keeping the CT death domain at bay for auto-inhibition" – does this sentence really express what it is intended express?

9)

Page 16: The authors state that all crystallographic structures of ASIC1 "lack the 25 NT and 64 CT residues and are non-functional". This is not true. Baconguis et al. (2012 and 2014) used a construct that lacked 13 NT residues and was functional and also the construct used by Yoder et al. (2018) was functional despite lacking 25 NT residues.

10)

Page 25: I think CFP-YFP is a positive (rather than negative) control. Isn't it?

11)

Page 34, legend to figure 2i: "Summary data for RIPK1 pulled down by the ASIC1a antibody." Does this data show the ratio RIPK1/ASIC1a? Please be more specific.

Reviewer #2:

Remarks to the Author:

In this paper, Wang and colleagues investigated potential contribution of N-terminal tail of ASIC1a to acidosis-induced neuronal injury. Previously the group has reported the discovery of a new channel-independent cell death function mediated by the C-terminus of ASIC1A (ref 2.). This study extends the previous finding and reported that the N-terminus contributes to an autoinhibition of the C-terminal "death motif". The key conclusion on the N-C terminal interaction is supported by three lines of evidence. 1) modeling based on Rosetta. 2) FRET based analysis of N- and C-terminal interaction 3) biochemical analysis of WT and mutant. Functional importance of this interaction is further examined in vitro (acidosis) and in vivo (ischemia). Besides the interaction between the tails, the authors further presented evidence to show that the interaction between N- and NSF is required for the effect of C-terminal-RIPK interaction. The findings suggest a potential new regulatory mechanism of ASIC1A in acidosis-induced neuronal death. There are some concerns in the current version:

1. One issue that needs to be considered is the pH sensitivity of YFP (e.g., Rekas JBC 277 pp. 50573–50578, 2002; Llopis Proc Natl Acad Sci USA. 95:6803–6808, 1995). pKa for EYFP is about 7.0 while some mutants such as citrin has a lower pKa around 6. One would expect a significant reduction in YFP emission starting at pH below 6.5. The slow reduction in FRET efficiency also appears to match well with the time needed for intracellular acidification following extracellular acidosis, which subsequently quenches YFP. While this predicts a reduction in YFP/CFP fluorescence

ratio, it is also a bit surprising that in Fig. 2C pH 6 had no effect on CFP-YFP FRET efficiency? Regardless of the CFP-YFP result, it seems that acid induced quenching of YFP signal alone can explain the small and slow reduction in FRET efficiency of the CFP-ASIC-YFP protein. Additional control experiments or alternative approach may be needed to strengthen this key piece of evidence.

2. ASIC1A has two short intracellular tails. Two fluorescent proteins on both ends seems bulky. Does adding two FPs interfere with the biogenesis or trafficking of the channel?

3. Some of the results in Figure 1 are not so easy to interpret. Fig 1c, why peptide 21-40 increased injury? Fig. 1f, Peptide 1-20 was protective at 5 & 10 uM, but had opposite (worsened injury) effect at 20 uM? What is the mechanism for the 20 uM effect? Whatever the mechanism is, the narrow concentration range may lead to issues on achieving optimal dose for in vivo therapy.

4. Potential interaction with NSF is interesting. NSF regulates vesicle trafficking and fusion. This raises a question of whether the effect of NSF, N-terminal, or RIPK is on ASIC intracellular trafficking or surface expression? This could potentially explain the reduction in acid currents in the mutants and the protein level (abnormal trafficking could lead to degradation).

5. The authors state that aa11-20 is likely the region interacting with NSF. Direct data support this claim seems needed. In addition, what would a peptide with WT(1-10)mut(11-20) do to survival and RIPK activation?

6. Most of the literature showed a good correlation between the magnitude of ASIC current and its potential in injury. This makes one wonder whether the channel-independent mechanism (ref #2) is a special mechanism that only kicks in under specific condition? In many neurons in brain, a large percentage of ASIC channels appears to be heteromeric channels. There are multiple studies showing that both ASIC2A and ASIC2B play indispensable role in acidotic neuronal injury. How would heteromerization with ASIC2A and ASIC2B alter the prediction, either the ASIC1A tail interaction or the channel-independent contribution to neuron death?

Minor issues:

1. Fig. 4d, it looks like NT1-20 still increased ASIC1a-NSF pull down in the pH 6 condition?

2. It may worth clarifying that, in the computer model, does the N-terminal tail of a specific subunit interacts with its own C-terminus, or the C-terminal of a neighboring subunit?

3. For spectral FRET, the exact parameters for spectrum imaging can be added to the methods section. For example, the spectral range captured and used for calculating the efficiency.

4. Despite the explanation by the authors, it is not so easy to fully digest the observation that the del20 mutant has greatly reduced expression but potent effect on inducing cell death.

Reviewer #3:

Remarks to the Author:

This research team has previously shown that ASIC1a can mediate necroptosis induced by acidification, via recruiting the kinase RIPK1 to its C-terminus. This process is independent of any ion currents passing through ASIC1a. Here they test the hypothesis that the intracellular N-terminal part of ASIC1a exerts under baseline conditions an autoinhibitory function by interacting with the C-terminus and preventing the interaction of the C-terminus with RIPK1. In support of

this hypothesis, the authors show first that a peptide corresponding to a part of the ASIC1a N terminus prevents acid-induced cell death, and show further that expression of a N-terminally truncated ASIC1a construct (thus a construct lacking the autoinhibitory sequence) induces cell death at physiological pH. An interaction partner of the ASIC1a N terminus, N-ethylmaleimide-sensitive fusion ATPase (NSF) is identified, which interacts with the N-terminus and promotes association of RIPK1 with the C-terminus. To understand the interactions between the ASIC1a N- and C-termini, the authors construct a structural model of the intracellular ASIC1a parts. They find that a stretch of Glu residues of the N-terminus is close to the RIPK1 interaction sequence present in the C-terminal sequence containing several Lys residues, suggesting electrostatic interactions between the termini. To describe possible changes in their conformation, the authors express ASIC constructs tagged with CFP and YFP at their N- and C-termini and carry out FRET experiments. These experiments indicate a slow decrease of FRET upon acidification, which is interpreted as a separation of the N- and C-termini. Finally, it is shown that a membrane-permeable version of the N-terminal peptide reduces the infarct volume in an ischemic stroke model. ASICs play an important role in the context of ischemic stroke, and their mode of action in this context has been a puzzle for a long time. This study is an interesting follow-up of the previous study, showing clear evidence for a protective role of an intracellular N-terminal ASIC1a sequence.

General comments

1. The FRET experiments are a central part of the study. It is however very difficult to imagine how changes in FRET between ASIC-attached CFP and YFP should represent changes in distance between the N-terminal EEE and the C-terminal KKK motif. First, CFP and YFP are quite big proteins of 35-40 kDa each. According to the description, an ASIC trimer would contain 3 CFP and 3 YFP molecules, all attached to the relatively short ASIC N- and C-termini. The reporter fluorophores are therefore much bigger than the peptide sequences whose movement they should record. How can the authors be sure that changes in FRET report differences in distance between the ASIC subunit N- and C-termini? Smaller fluorophores need to be used for this assay. The EEE motif is located at the very beginning of the ASIC1a N-terminus, while the motif with the K residues is more in the center of the C-terminus. To be able to follow the interaction between these motifs, it would be good to place the fluorophore of the C-terminus close to the RIPK1 interaction motif, and not at its end. Based on the structural model of the intracellular part of the ASIC trimer, it needs to be shown how about these CFP and YFP molecules would be positioned relative to the ASIC channel.
2. For some of the approaches, the methods and conditions are not sufficiently well (or in some cases not at all) presented. For the approaches, this concerns the modeling and testing of the models, which is not explained, and the constructs for the FRET experiments. Besides providing information on the construction, testing and selection of the models, the quality, number and size of the structural model images in the manuscript needs to be improved, to allow the reader to understand the predictions. Are the CFP and YFP attached to the ends of full length ASIC1a, or to truncated constructs, as Suppl. Fig. 2a might indicate? In the context of the CFP and YFP constructs, the term "concatemer" is used. Does this apply to the CFP-YFP construct, or to any ASIC concatemers? The conditions and equipment for the FRET experiments need to be described. Indicate how the bands of Western blots were quantified. For the quantitative analysis of band intensities (Figs. 2i, 6f, 6g) indicate to which condition the signal intensities are normalized. For the following experiments, the conditions should be better defined: Fig. 2d-e: what was the pH of incubation? Fig. 6h, indicate the Nec-1 and NT1-20 and NT1-20E/A peptide concentrations used.
3. It is reported in this study that some of the truncation mutants, and the EEE-AAA mutant show very low expression. In the context of the elucidation of the mechanism of the involvement of ASIC1a in necroptosis, it is important to know whether the channels need to be present at the plasma membrane in order to mediate necroptosis. It is therefore critical to provide data on the cell surface expression of these mutants.

Specific points

1. Are RIPK1 and NSF endogenously expressed in CHO cells?
2. The observation that the NT1-20 peptide has a protective effect at 5 and 10 microM, but not at

20 microM, is somewhat confusing. How could the decrease in protection at higher peptide concentrations be explained? This should be discussed in the manuscript.

3. How is it explained that PcTx1 prevents in some conditions the ASIC1a-mediated necroptosis? By shifting the pH dependence of inactivation, PcTx1 affects ASIC opening. How would this affect a signaling that does not depend on ion permeation? Demonstration of prevention or inhibition of the conformational changes in the intracellular ASIC parts by PcTx1 would be a strong argument in favor of the proposed mechanisms of necroptosis.

4. Page 13, bottom, "Therefore, disrupting..." Since the NSF binding site is on the N-terminal, it is also quite likely that this mutation could also interfere with NSF binding. How can the authors exclude such a possibility?

5. There is clear evidence for a role of NSF in contributing to necroptosis. It seems however that this effect could be independent of the N- and C-terminus interaction. The authors should clearly develop their arguments for a role of NSF in inhibiting the interaction between the ASIC1a N- and C-termini, and, if this is not possible, present their conclusion ("Plausibly, by binding to ASIC1a NT, NSF helps keep the CT death motif free to interact with RIPK1, ...") as hypothesis.

6. P.16, lower paragraph, "Unfortunately, all these structures lack the..". It is not true that all these constructs are non-functional. Some of them were shown to be functional.

7. In the legend to Figs. 2d-e it is indicated that the PI staining was carried out after 24 h. Does this mean 24h after the transfection? Please clarify. The inhibitors were added 1h before the PI staining in these experiments. Assuming that the ASIC constructs were expressed a few hours after transfection, how can it be explained that exposure to the inhibitors in the last hour was sufficient to prevent cell death?

8. Fig. 2g, the current amplitudes need to be indicated as absolute current amplitude or current densities, not as normalized values.

Reviewer #1:

Wang et al. previously showed (E-Life, 2015) that a direct interaction of the C-terminus of acid-sensing ion channel (ASIC) 1a with receptor-interacting protein kinase (RIP) 1 leads to phosphorylation of RIP1 and ensuing necroptosis. In the present manuscript, they extend these findings showing that the interaction of the ASIC1-C-terminus with RIP1 is prevented at neutral pH by a tight interaction of the ASIC1 C-terminus with the ASIC1 N-terminus. This interaction is slowly released (with a time course of a few minutes) by extracellular acidification, allowing association of the C-terminus with RIP1. Moreover, they show that the N-ethylmaleimide sensitive factor (NSF) interacts with the free N-terminus of ASIC1a to prevent its re-association with the C-terminus. These protein interactions are all independent of ions permeating through the ASIC1 pore. The authors come up with a model in which they claim that conformational changes triggered by extracellular acidification lead to the slow dissociation of N- and C-termini of ASIC1, followed by their association with NSF and RIP1, respectively, which would finally induce necroptosis. Thus, they propose a model in which ASIC senses protons not just to open its ion pore but to trigger intracellular downstream events that are independent of its ion channel function. Although to my knowledge the original study published in E-Life has not yet been reproduced by others and although some mechanistic details remain unclear, this is a highly original study which opens up a completely new view on the role of ASIC1a in ischemic neuronal death. I am convinced that it will inspire a lot of follow-up studies. Therefore, this manuscript is of great interest to the field of ASICs and of basic mechanisms in ischemic neuronal death. It also has a clear translational relevance.

Response: We thank the reviewer for the positive comments.

I have a few comments to further improve this manuscript:

Major comments:

1) The authors claim that conformational changes in ASIC1a triggered by extracellular acidification induce the slow dissociation of N- and C-terminus. The evidence for this hypothesis is rather indirect, however. An alternative explanation would be, for example, that the extracellular acidification leads to a slow intracellular acidification which then leads to the dissociation of N- and C-termini. Relatively easy ways to discriminate between these two possibilities would be, first, to follow the decrease in pH_i by pH-imaging in CHO cells and in parallel to measure FRET like in Fig. 2b. Is the time course of both events similar? Second, the authors could try to reduce intracellular pH by other means (for example by blocking NHEs). Third, they could use mutants of ASIC1a that are no longer competent for conformational changes, for example due to engineered cysteine bonds. Such a mutant has been described by the Kellenberger group (E235C/Y389C; Gwiazda et al. JBC 2015). Is such a mutant still able to induce necroptosis, for example in CHO cells? Although these experiments will not provide a definite answer, they could help support the conclusion of an extracellular conformational change transmitted to the intracellular termini.

Response: We thank Reviewer #1 for the comments and constructive suggestions. We have followed the reviewer's suggestion with the following experiments. First, we

measured pH_i by BCECF imaging of CHO cells. With the change of extracellular pH from 7.4 to 6.0, the time course of intracellular acidification (Fig. R1a) appears to match that of the FRET decrease of CFP-ASIC1a-YFP (Fig. 2b of the ms). This would be in line with the reviewer's suggestion that intracellular acidification may underlie the dissociation of ASIC1a-CT from its NT. We also reduced pH_i , as the reviewer suggested, by blocking NHEs with 10 μ M cariporide and repeated the experiments. Cariporide induced a slow decrease in pH_i and in parallel with the pH_i decrease, the YFP/CFP fluorescence intensity ratio (F_{525}/F_{482} nm emission with 405 nm excitation) of CFP-ASIC1a-YFP also decreased (Fig. R1b, R1c), supporting the idea that intracellular acidification alone can induce the dissociation of ASIC1a-CT from its NT. However, it is hard to assess that under these experimental conditions whether or not the cariporide treatment also affected the local pH near the extracellular domains of ASIC1a. Presumably, such a pH_e change, if occurred, would be too slow to induce detectable ASIC1a current (Wang et al. eLife 2015)⁵ and too local to be measurable by monitoring the pH_e in the buffer. More surprisingly, we found that the near membrane pH_i drop, as monitored by ASIC1a-YFP taking advantage of proton quenching of the YFP fluorescence signal, in response to extracellular acidification occurs much faster (Fig. R4b) than the average pH_i change revealed by BCECF. Since the near membrane pH_i change is more relevant to the membrane localized ASIC1a, the mismatch between the kinetics of pH_i and FRET changes directly underneath the membrane suggests that the slow dissociation of the ASIC1a-CT and NT might be intrinsic to the channel complex. Clearly, this represents a complex issue that requires a lot of future work to resolve.

Finally, as the reviewer suggested, we expressed the E235C/Y389C mutant of ASIC1a (ASIC1a-E235C/Y389C) in CHO cells and measured FRET change during the treatment with the pH 6.0 solution. The FRET (YFP/CFP fluorescence intensity ratio) response of ASIC1a-E235C/Y389C to extracellular acidosis was markedly delayed and reduced as compared to the wild type ASIC1a (Fig. R1d). Notably, the mutated sites of ASIC1a-E235C/Y389C are extracellular and the mutant is not completely non-functional, retaining activity at $pH < 4$ and exhibiting steady-state desensitization at $pH < 7$.¹ Thus, this result demonstrates a close relationship between the extracellular acid-induced FRET decrease of CFP-ASIC1a-YFP and conformation change of ASIC1a protein, ruling out the possibility that the observed FRET change was entirely due to pH_i effect on the fluorescence proteins (see also our response to Reviewer #2, point 1). Please see page 8, lines 174 to 177 and page 9, lines 179 to 181 for the revised text. Please also refer to revised Fig. 2b, 2c and the legend for details (see page 36, lines 861 to 869). Taken together, the results from these new experiments indicate that although the dissociation of ASIC1a termini under acidosis condition could result from intracellular acidification, the involvement of extracellular acidification cannot be completely ruled out. We have corrected the description as "Rather, this acidotoxic effect involves an acidosis-evoked complex formation between ASIC1a and receptor-interacting serine/threonine-protein kinase1". Please see page 3, lines 59 to 61.

Fig. R1. Intracellular acidification leads to the dissociation of ASIC1a-CT from its NT. (a) The time course of intracellular pH (pH_i) changes measured by BCECF imaging with the change of extracellular pH from 7.4 to 6.0 in CHO cells. (b) 10 μ M cariporide (NHE inhibitor) induced a slow decrease of pH_i in CHO cells. (c) 10 μ M cariporide decreased the YFP/CFP fluorescence intensity ratio of CFP-ASIC1a-YFP in FRET analysis. (d) Changes in YFP/CFP fluorescence intensity ratio of the E235C/Y389C mutant of ASIC1a (ASIC1a-E235C/Y389C) in CHO cells during the treatment with the pH 6.0 solution.

2) The statistical analysis can be improved at several instances and more information on statistical analysis needs to be provided:

a) Please mention unequivocally whether assays (for example the death assays) included technical replicates or only biological replicates and what a biological replicate exactly meant. On page 26 the authors state “All death assays were performed with more than four repeats each time” As some graphs show only four symbols, I guess what is really meant is “All death assays were performed with at least four repeats each time”?

Response: Biological replicates are parallel measurements of biologically distinct samples, e.g. cells from separated dishes and experiments carried out in the same manner but on different days. Only biological replicates are included in this manuscript. We have modified the sentence as “All death assays were performed with more than four biological replicates” (please see page 29, lines 653 to 654).

b) Was the experimenter really blind to group allocation for all assays as is mentioned in the Reporting Summary? Please state this explicitly also in the Methods section under “Statistical analysis”.

Response: Yes, the experimenters were blind to group allocation for all assays as is

mentioned in the Reporting Summary. We have also indicated this in the Methods section under “Statistical analysis” as suggested “During data collection, the experimenters were blind to group allocation to avoid experimenter bias” (please see page 30, lines 677 to 678).

c) It appears that in different graphs “control” means something different. In some graphs at least it is a control peptide. Which peptide has been used as a control? The authors need to mention its sequence and concentration. Please state in each case explicitly what the control was.

Response: We apologize for the misleading data presentation. To make it clear, we followed the reviewer’s suggestion and used different term in each case in the revised manuscript: Ctrl in Fig. 1 is TAT alone at 10 μ M, refers to control peptide (see page 36, line 854); Ctrl in Fig. 3, now renamed as “Contra”, refers to the contralateral cortices of the same mice subject to intraluminal middle cerebral artery occlusion (see page 11, line 238), and Ctrl in Fig. 5, now renamed “Scrm”, refers to shRNA with the scrambled sequence (CTTAAGGTTAAGTCACTCT, see page 30, line 661). All corresponding corrections are revealed in the figures and highlighted in the revised manuscript.

d) Please explicitly state in figure legends (or in the text), which type of statistical analysis has been done (t test or ANOVA). Multiple t-tests using the same control need to be adjusted for multiple comparisons (for example, by Bonferroni correction).

Response: We thank the reviewer for the kind reminder and have stated the statistical method explicitly in the legend of each figure. We used t test to assess the difference between two groups, and One-way ANOVA and Two-way ANOVA to assess the difference between multiple groups.

e) Experiments reported in figures 3c-e, 4c-d, and 5c (three repetitions) should be quantified and the quantitative summary data should be shown.

Response: We thank the reviewer for the comment and have added the statistical results of Fig. 3c, 3e and 3g, Fig. 4c and 4e, and Fig. 5c in the revised manuscript.

f) Not always does the statistical analysis, which has actually been done, exactly support what is said in the text. For example, on page 13 it is said that “shRNA knockdown of NSF abolished acidosis-induced neuronal death, as shown by PI staining”. But the statistics compares the difference in acidosis-induced cell death between two conditions. Acidosis actually seems to still induce cell death also in the shRNA group, although to a strongly reduced amount. (if this is the case, a statement like “shRNA knockdown of NSF attenuated acidosis-induced neuronal death” would be more appropriate). Another example: while on page 32 it is said that “CP-1-3 did not induce death”, figure 1b shows significantly reduced viability in the presence of CP-1-3. Furthermore, still on page 32 it is said that “pH 6.0-induced cell death was significantly reduced by the pretreatment of NT1-20 but not control peptide.” But the statistics compares NT1-20 vs control (and not NT1-20 and control vs. vehicle).

Response: We apologize for the imprecise statements. We agree with the reviewer that “attenuated” is more appropriate than “abolished” to describe the effect of NSF shRNA on acidosis-induced neuronal death and have made the change as suggested (see page 14, line 302 and page 39, line 925). We have also corrected the description for Fig. 1b as “Compared to CP-1-2, CP-1-3 induced less and the membrane impermeable CP-1-2S induced no death” (see page 5, lines 98 to 100 and page 35, lines 835 to 837), and “pH 6.0-induced cell death was significantly reduced by the pretreatment of NT₁₋₂₀, but not the half-split NT fragments, NT₁₋₁₀ and NT₁₁₋₂₀” on page 35, lines 844 to 845 in the revised manuscript.

g) The authors should consider providing exact P values (rather than, for example, P < 0.01).

Response: We thank the reviewer for the comment and have provided the exact p values in Supplementary Table 2 in the revised manuscript.

h) In figure 5a, a paired t test has been performed (pages 37/38). Since I doubt that identical neurons have been transfected with control shRNA and with NSF shRNA, the use of a paired t test is not appropriate in this case.

Response: We thank the reviewer for the careful reading. We respectfully disagree with the reviewer on the appropriate statistical method for figure 5a. For quantification, we first normalized the expression of NSF in the two groups to that of GAPDH on the same PVDF membrane and then made a ratio of NSF levels in the NSF shRNA group vs. control shRNA group. Therefore, we believe that paired t test is appropriate here.

3) In order to facilitate reproduction of the study, please mention the amount of neurons and the weight of brain tissue used for co-immunoprecipitation (Methods, page 23). Also, please clearly state how the standard medium (SS) was buffered to pH 6.0. Was the concentration of bicarbonate adjusted? Was another buffer (HEPES?) present (Methods, page 25)? Finally, please mention how many peptide sequences were identified by MALDI-TOF. (Page 34: “18 peptide sequences were found to match NSF” - 18 out of how many?) Any other hits that turned up several times?

Response: We thank the reviewer for the suggestion. We have revised the method of co-immunoprecipitation with greater details as suggested (see page 25, lines 549 to 550). As for the standard external solution (SS), we have stated it clearly in page 26, lines 583 to 585 “The standard external solution (SS) contained: 150 mM NaCl, 5 mM KCl, 1 mM MgCl₂, 2 mM CaCl₂, 10 mM glucose and 10 mM HEPES buffered to various pH values with Tris-base or HCl.” There is no bicarbonate added in these solutions.

In Fig. 3b, 18 peptide sequences (out of 39) were found to match NSF (see page 37, line 893). We did not find many peptide species because only a thin band was cut and used for the Mass-Spec.

Minor comments:

1) Page 5: The numbering of the CP-1 peptide (amino acids 463-483) is correct but does not correspond to the numbering published in the E-Life paper. Perhaps the authors could clarify this discrepancy in a short note somewhere in the Methods.

Response: As the reviewer noted, the numbering of the CP-1 peptide in the original eLife paper had an error. We have published a corrigendum about this point in eLife 2016, 22; 5:e14128.

2) Page 6: any idea why the NT1-20 peptide lost its protective effect at a high concentration (20 microM)?

Response: We thank the reviewer for this question. As all the peptides used contain HIV-1 TAT protein transduction domain to help penetration of cell membrane and TAT has been reported to induce redox-related inflammatory responses both in vitro and in vivo^{2,3}, we considered the toxicity of TAT as a possible reason for the lost protective effect of NT₁₋₂₀ at 20 μ M. To test this possibility, we examined the effect of control peptide (TAT only) on acid-induced LDH release in neurons. Indeed, at 20 μ M, but not 5 and 10 μ M, TAT itself increased the acid-evoked LDH release (Fig. R2, also Supplementary Figure 1c of the revised ms), showing that TAT peptide can exacerbate cell death under acid treatment. This nonspecific toxicity of TAT may account for the lack of effect of NT₁₋₂₀ at the high concentration. We have updated this part in the revised manuscript accordingly. Please see page 6, lines 123 to 126. Please also see the revised Supplementary Fig. 1c and the legend for details (see page 41, lines 984 to 987).

*Fig. R2. High concentration (20 μ M) of the TAT alone peptide increased acid-induced LDH release in neurons. n=4, ***p < 0.001 vs. corresponding pH 7.4; ### p < 0.001 vs. vehicle (Veh) in pH 6.0.*

3) Fig. 1c: any idea, why NT21-41 increased (rather than decreased) LDH release? If the increase was significant, this should be indicated on figure 1c.

Response: It is an interesting phenomenon that pretreatment with NT₂₁₋₄₁ significantly increased CP-1-2 induced LDH release. We do not know exactly the mechanism. However, we confirmed that the RIPK1 signaling pathway is not involved here because the increased LDH release by NT₂₁₋₄₁ was not inhibited by Nec-1 (Fig. R3).

*Fig. R3. LDH release assay for viability of neurons treated with NT₂₁₋₄₁ without or with Nec-1. NT₂₁₋₄₁ (10 μ M) significantly induced LDH release and this effect was not inhibited by Nec-1 (20 μ M). n=4, ***p < 0.001 vs. corresponding control peptide (Ctrl).*

4) Fig. 2a: the difference between the two conformations is difficult to see in the blow-up.

Response: Thanks for bringing up this point. We have revised the picture so that it can better reflect the two conformations. Please see revised Fig. 2a and the legend for details (see page 36, lines 857 to 860).

5) Page 13 and page 36: Please change “positively changed lysine” to “positively charged lysine”.

Response: Thanks for pointing out this inadvertent error. We have corrected it.

6) Page 13: Does figure 6b really depict FRET efficiency? On page 8, it is explained that spectra FRET is needed to measure C.

Response: No, Fig. 6b depicts YFP/CFP intensity as in Fig. 2b, not FRET efficiency. We have stated it clearly in the revised text (see page 15, lines 323 to 325 and page 40, lines 949 to 952). Only Fig. 2c (of the original ms) was obtained from spectra FRET. For live cell monitoring of acidosis-induced changes in NT-CT interaction of ASIC1a, we used crude measurements of acceptor (YFP) emission (525 nm) with the donor (CFP) excitation (405 nm) in order to collect detailed kinetic information. We also measured donor (CFP) emission (482 nm) simultaneously and used the emission ratio (YFP/CFP) to represent FRET. Admittedly, the fluorescence protein (FP)-based FRET analysis in live cells has some inherited pitfalls as explained before⁴. Therefore, in order to validate the observed changes from crude YFP/CFP intensity analysis, we also performed spectra FRET analysis to determine FRET efficiency of the same constructs (CFP-ASIC1a-YFP,

with CFP-YFP as the control) in cells bathed in pH 7.4 and pH 6.0 solutions. Spectra FRET allows correction of multiple problems associated with the YFP/CFP intensity measurement, such as the cross-talk, bleed-through, variable expression levels of FPs among individual cells, and certain non-specific FRET signals⁴. Specifically, we made independent calibration for cross-talk and bleed-through signals of YFP and CFP, respectively, at pH 7.4 and pH 6.0 in order to correct for the fluorescence property changes induced by low pH. Under the same experimental conditions and with the change of pH_e from 7.4 to 6.0, we observed significant decrease in FRET efficiency only in CFP-ASIC1a-YFP and CFP-ASIC1a-HIF-YFP, but not CFP-YFP (Fig. 2d of the revised ms). These data help validate the results from the crude YFP/CFP intensity measurement showing the specific involvement of ASIC1a in the acid-induced change in CFP-ASIC1a-YFP FRET. Together with the new negative control data on CFP-ASIC1a-E235C/Y389C-YFP and CFP-ASIC1a-YFP expressing cells treated with PcTX1 and NSF shRNA (new Figs. 2b, 2c, 5e, and 5f of the revised ms), our data collectively demonstrate an acid-induced separation of ASIC1a-NT from its CT that requires conformational (but not ionotropic) signaling of ASIC1a and the newly identified chaperon, NSF, as a critical component of the conformational signaling.

7) If I see it correctly, in the main text there is no reference to Supplementary figure 4.

Response: We have included the reference to previous Supplementary Figure 4 in the main text (see page 18, lines 394).

8) Page 14: “are crucial for keeping the CT death domain at bay for auto-inhibition” – does this sentence really express what it is intended express?

Response: We apologize for the misunderstanding of this sentence. We have revised it as the following: “These results strongly suggest that the negatively charged E⁶EEE⁹ at ASIC1a-NT are crucial for interaction with the CT death domain, creating auto-inhibition under resting conditions” (see page 15, lines 329 to 331).

9) Page 16: The authors state that all crystallographic structures of ASIC1 “lack the 25 NT and 64 CT residues and are non-functional”. This is not true. Bacongus et al. (2012 and 2014) used a construct that lacked 13 NT residues and was functional and also the construct used by Yoder et al. (2018) was functional despite lacking 25 NT residues.

Response: The reviewer is correct. Based on the literature, ASIC1a lacking 25 NT residues was functional. We have adjusted this statement in the revised manuscript (see page 18, line 390).

10) Page 25: I think CFP-YFP is a positive (rather than negative) control. Isn't it?

Response: CFP-YFP is a negative control here because the two fluorescent domains do not dissociate under acidosis treatment.

11) Page 34, legend to figure 2i: “Summary data for RIPK1 pulled down by the ASIC1a antibody.” Does this data show the ratio RIPK1/ASIC1a? Please be more specific.

Response: Yes, figure 2i (now 2j) shows the ratio RIPK1/ASIC1a and ASIC1a was used as an internal control. To make it clear, we have re-named the Y axis as “Fold change of RIPK1/ASIC1a” in the revised Fig. 2j.

Reviewer #2:

In this paper, Wang and colleagues investigated potential contribution of N-terminal tail of ASIC1a to acidosis-induced neuronal injury. Previously the group has reported the discovery of a new channel-independent cell death function mediated by the C-terminus of ASIC1A (ref 2.). This study extends the previous finding and reported that the N-terminus contributes to an autoinhibition of the C-terminal “death motif”. The key conclusion on the N-C terminal interaction is supported by three lines of evidence. 1) modeling based on Rosetta. 2) FRET based analysis of N- and C-terminal interaction 3) biochemical analysis of WT and mutant. Functional importance of this interaction is further examined in vitro (acidosis) and in vivo (ischemia). Besides the interaction between the tails, the authors further presented evidence to show that the interaction between N- and NSF is required for the effect of C-terminal-RIPK interaction. The findings suggest a potential new regulatory mechanism of ASIC1A in acidosis-induced neuronal death. There are some concerns in the current version:

We are very grateful to Reviewer #2 for the comments. We summarized the answers to the questions raised as follows.

1. One issue that needs to be considered is the pH sensitivity of YFP (e.g., Rekas JBC 277 pp. 50573–50578, 2002; Llopis Proc Natl Acad Sci USA. 95:6803–6808, 1995). pKa for EYFP is about 7.0 while some mutants such as citrin has a lower pKa around 6. One would expect a significant reduction in YFP emission starting at pH below 6.5. The slow reduction in FRET efficiency also appears to match well with the time needed for intracellular acidification following extracellular acidosis, which subsequently quenches YFP. While this predicts a reduction in YFP/CFP fluorescence ratio, it is also a bit surprising that in Fig. 2C pH 6 had no effect on CFP-YFP FRET efficiency? Regardless of the CFP-YFP result, it seems that acid induced quenching of YFP signal alone can explain the small and slow reduction in FRET efficiency of the CFP-ASIC-YFP protein. Additional control experiments or alternative approach may be needed to strengthen this key piece of evidence.

Response: Yes, we agree with the reviewer that YFP is sensitive to pH and the decrease in the apparent CFP-ASIC1a-YFP FRET could result from the quench of YFP by the acidic pH_i. To address this point, we performed additional control experiments for the effects of pH on fluorescence intensities of CFP-ASIC1a, ASIC1a-YFP, and CFP-ASIC1a-YFP. Indeed, a switch of the extracellular pH from 7.4 to 6.0 drastically reduced the fluorescence intensity of ASIC1a-YFP (Fig. R4b, also Supplementary Figure 2e of the revised ms). Interestingly, the rate of ASIC1a-YFP fluorescence decrease is much faster

than that of the global cytosolic pH change measured using BCECF (Fig. R1a). Please see page 9, lines 181 to 201 for the revised text. Please also refer to revised Supplementary Figures 2d-e and the legend for details (see page 42, lines 1004 to 1008). This may suggest that the membrane-targeted YFP is better at detecting the near-membrane pH_i than the cytoplasmically loaded dye. If this is true, then the extracellular acid-induced pH_i change near the plasma membrane localized ASIC1a may be much faster than that revealed by the pH_i measurement using BCECF (Fig. R1a).

Despite the dramatic change in ASIC1a-YFP, the fluorescence intensity of CFP-ASIC1a did not change much (Fig. R4a). Moreover, using 488 nm excitation and 525 nm emission to assess YFP in CFP-ASIC1a-YFP, we found that the response to pH 6.0 treatment was markedly different from that of ASIC1a-YFP (Fig. R4b, also Supplementary Figure 2e of the revised ms). Instead of a rapid drop, as in the case of ASIC1a-YFP, CFP-ASIC1a-YFP displayed a slow, gradual, decrease, reminiscent of the change pattern of YFP/CFP emission ratio induced by the pH 6.0 solution (Fig. 2b of the ms). Plausibly, the presence of CFP in close vicinity prevents proton quenching of YFP fluorescence. This could explain why the CFP-YFP concatemer was completely insensitive to the acidosis challenge (Fig. 2d of the revised ms). Then, the change in the YFP signal in CFP-ASIC1a-YFP should really just reflect the dissociation of CFP from the YFP in this construct, making YFP assessable to quenching by the protons.

To further validate that the recorded FRET change was due to the conformational change of ASIC1a, we expressed the E235C/Y389C mutant of ASIC1a (CFP-ASIC1a-E235C/Y389C-YFP) in CHO cells and measured the acid-induced FRET change. This mutant exhibits markedly reduced sensitivity to acidic pH, requiring pH < 4 for activation and pH < 7 for steady-state desensitization¹. The FRET response of CFP-ASIC1a-E235C/Y389C-YFP to pH 6.0 treatment was markedly delayed and reduced as compared to the wild type ASIC1a (see Fig. R1d, also Fig. 2b the revised ms). Please see page 8, lines 174 to 177. In addition, the low pH-induced FRET change of CFP-ASIC1a-YFP was significantly inhibited by blocking ASIC1a with PcTX1 and siRNA-mediated knockdown of NSF, a newly identified chaperone that specifically regulates conformational signaling of ASIC1a (Fig. R10, also Figs. 2b and 5e of the revised ms). Please see page 9, lines 177 to 181 and page 14, lines 298 to 301. These results demonstrate a close relationship between the acidosis-induced FRET decrease of CFP-ASIC1a-YFP and the conformation change of ASIC1a protein, ruling out the possibility that the observed FRET change was entirely due to pH_i effect on the fluorescence proteins (see also our response to Reviewer #1, major point 1 and minor point 6). Please refer to revised Figs. 2b, 2c, 5e, 5f and the legend for details (see page 36, lines 860 to 869 and page 39, lines 928 to 932).

Collectively, our original data and the new data present here, as well as our results of De novo Rosetta modeling of ASIC1a (Fig. 2a), strongly suggest that extracellular acidosis induces a dissociation between ASIC1a-NT and its CT, although we cannot fully exclude that other interactions may also be affected by such a treatment.

Fig. R4. Effects of decreasing pH_e on fluorescence intensities of CFP-ASIC1a, ASIC1a-YFP, and CFP-ASIC1a-YFP. (a) The fluorescence intensity of CFP-ASIC1a did not change much in response to the pH 6.0 treatment. (b) pH 6.0 treatment drastically reduced the fluorescence intensity of ASIC1a-YFP, but not the fluorescence intensity of YFP in CFP-ASIC1a-YFP (using 488 nm excitation and 525 nm emission).

2. ASIC1A has two short intracellular tails. Two fluorescent proteins on both ends seems bulky. Does adding two FPs interfere with the biogenesis or trafficking of the channel?

Response: We thank the reviewer for bringing up this concern. To address this issue, we measured the current of CFP-ASIC1a-YFP. The results from this experiment suggest that adding the two FPs reduced acid-induced ASIC1a current, with also a decreased rate of desensitization, as compared to having just one FP (ASIC1a-YFP) (Fig. R5). However, the time courses for the activation and desensitization of the current are still much faster than that of acid-induced FRET decrease, i.e. dissociation between the N and C-termini of ASIC1a. It is possible that the bulky CFP and YFP tags at the cytoplasmic ends make the conformational change more sluggish. However, this should not alter our overall conclusion about the presence of an N- to C-interaction at the cytoplasmic termini of ASIC1a and its dissociation upon acid stimulation. Our data on N-terminal deletion mutants, N-terminal peptides, NSF binding, and Rosetta modelling all support this major conclusion.

*Fig. R5. Adding CFP and YFP to the cytoplasmic termini of ASIC1a reduced acid-induced ASIC1a current. Shown are representative current traces at -60 mV with whole-cell patch-clamp techniques (a) and summary data for peak currents (b) of ASIC1a-YFP and CFP-ASIC1a-YFP expressed in CHO cells. $n=6-7$, $***p<0.001$ vs. ASIC1a-YFP.*

3. Some of the results in Figure 1 are not so easy to interpret. Fig 1c, why peptide 21-40 increased injury? Fig. 1f, Peptide 1-20 was protective at 5 & 10 μ M, but had opposite (worsened injury) effect at 20 μ M? What is the mechanism for the 20 μ M effect? Whatever the mechanism is, the narrow concentration range may lead to issues on achieving optimal dose for in vivo therapy.

Response: We thank the reviewer for pointing out these important questions, which are also raised by Reviewer #1. Please refer to our responses to Reviewer #1's Minor points 2 and 3). Briefly, NT₂₁₋₄₀ caused a RIPK1-independent toxicity (Fig. R3) and 20 μ M NT₁₋₂₀ lost the protection because of the toxicity from the TAT tag (Fig. R2, also Supplementary Figure 1c of the revised ms), which was used to facilitate cell penetration. Future work should focus on identifying alternative peptide sequences to facilitate cell penetration.

4. Potential interaction with NSF is interesting. NSF regulates vesicle trafficking and fusion. This raises a question of whether the effect of NSF, N-terminal, or RIPK is on ASIC intracellular trafficking or surface expression? This could potentially explain the reduction in acid currents in the mutants and the protein level (abnormal trafficking could lead to degradation).

*Response: We completely agree with the reviewer on this and considered it an excellent idea to test the effects of NSF, N-terminal deletion, or RIPK1 on ASIC1a intracellular trafficking and surface expression. We have shown that knocking down NSF by shRNA has no effect on the expression of ASIC1a and acid-induced current (Supplementary Figure 3). Here, we carried out additional studies to address the effects of NT deletion and blocking RIPK1 with Nec-1. The results from these new experiments show that neither NSF shRNA, nor NT deletion, nor Nec-1 could affect the surface/total ratio of ASIC1a (Fig. R6a, b and c), despite the dramatic decrease in the overall level by the NT deletion (Fig. 2i and j of the revised ms), suggesting no change per se in trafficking or surface delivery. We have also added the method for surface biotinylation assay in the revised text. **Please see page 28, lines 624 to 634.***

Fig. R6. The effects of NSF, NT deletion, and RIPK1 on ASIC1a intracellular trafficking and surface expression. Neither NSF shRNA (a), nor NT deletion (b), nor RIPK1 inhibitor Nec-1 (c) affected the surface/total ratio of ASIC1a. (a) is also shown in Supplementary Figure 3a.

5. The authors state that aa11-20 is likely the region interacting with NSF. Direct data support this claim seems needed. In addition, what would a peptide with WT (1-10) mut (11-20) do to survival and RIPK activation?

Response: We thank the reviewer for bringing up this question. Based on our co-IP data that deleting aa1-10 ($\Delta 1-10$) did not abolish acid-induced increase in NSF-ASIC1a association, but deleting aa1-20 ($\Delta 1-20$) did (Fig. 4c), we speculated that residues within aa11-20 must be critical for the interaction with NSF. However, $\Delta 1-10$ nonetheless showed less co-IP with NSF than the full-length ASIC1a, suggesting that some residues in aa1-10 may also contribute to the binding. We agree with the reviewer that more work is needed to clearly define the molecular interaction between ASIC1a and NSF, which is the priority of our future study. Here we modified the sentence as “based on the above data, we speculate that residues in aa11-20 contribute critically to ASIC1a interaction with NSF” (see page 13, lines 282 to 283) in the revised manuscript.

As for the effect of peptide with WT (1-10) mut (11-20) on survival and RIPK activation, we do not think that it has the same protective effects as peptide NT₁₋₂₀. Additional experiments are needed to substantiate this point in the future.

6. Most of the literature showed a good correlation between the magnitude of ASIC current and its potential in injury. This makes one wonder whether the channel-independent mechanism (ref #2) is a special mechanism that only kicks in under specific condition? In many neurons in brain, a large percentage of ASIC channels appears to be heteromeric channels. There are multiple studies showing that both ASIC2A and ASIC2B play indispensable role in acidotic neuronal injury. How would heteromerization with ASIC2A and ASIC2B alter the prediction, either the ASIC1A tail interaction or the channel-independent contribution to neuron death?

Response: We thank the reviewer for this insightful comment. We have previously shown that neuronal necrosis induced by acidosis was dependent on treatment duration, ASIC1a function, but not ASIC1a current (Wang et al., eLife 2015)⁵. We speculate that the channel-independent mechanism of ASIC1a may mainly function under conditions of slow and persistent tissue acidosis.

To test whether NT₁₋₂₀ has a protective effect on heteromeric ASIC1 channels, we co-transfected ASIC1a and ASIC2a or ASIC1a and ASIC2b in CHO cells, and measured LDH release induced by the pH 6.0 solution. We found that NT₁₋₂₀ reduced LDH release in both groups (Fig. R7), supporting that the same conductance-independent mechanism is also at work for cell death mediated by heteromeric ASIC1/2 channels. Note, NT₁₋₂₀ does not alter the acid-evoked current (Fig. 1i of the manuscript).

Fig. R7. Treatment with NT₁₋₂₀ reduced cell death mediated by heteromeric ASIC1/2 channels. (a) Summary data for PI staining of CHO cells co-transfected with ASIC1a and ASIC2a. (b) Summary data for PI staining of CHO cells co-transfected with ASIC1a and ASIC2b. The CHO cells were treated with either pH 7.4 or pH 6.0 SS for 1 hr and then returned to the normal culture medium for 24 hr before PI staining. NT₁₋₂₀ was added 1 hr before pH 6.0 treatment.

Minor issues:

1. Fig. 4d, it looks like NT1-20 still increased ASIC1a-NSF pull down in the pH 6 condition?

Response: Yes. Although NT₁₋₂₀ may be able to sequester NSF, it does not appear to be able to completely disrupt acidosis-induced ASIC1a-NSF interaction (Fig. 4e-f of the revised ms). We believe that NT₁₋₂₀ mainly exerts its neuroprotective effect through binding to ASIC1a-CT and thereby occluding RIPK1 from interacting with ASIC1a.

2. It may worth clarifying that, in the computer model, does the N-terminal tail of a specific subunit interacts with its own C-terminus, or the C-terminal of a neighboring subunit?

Response: We thank the reviewer for the helpful comment and agree with the reviewer that it is of interest to distinguish whether the NT-CT interactions of ASIC1a are inter-subunit or intra-subunit or both. In the Rosetta model, the interactions were modeled for N- and C-termini of the same subunit (please see page 15, lines 316 to 318). In our FRET experiments, CFP and YFP were fused to NT and CT, respectively, in each subunit of the ASIC1a trimer, and as a result, the interactions could be either inter-subunit or intra-subunit. To address the reviewer's question, we made use of a tandem concatemer which has three ASIC1a subunits connected as one polypeptide. To examine inter-subunit interaction, we tagged a CFP at the N terminus and a YFP at the C terminus; to test intra-subunit interaction, we tagged a CFP at the N terminus and inserted a YFP after the first subunit (Fig. R8a). The FRET experiments showed that both interactions were disrupted by extracellular acidosis as in the case of CFP-ASIC1a-YFP (Fig. R8b and c). These results suggest that both inter- and intra-subunit interactions may occur between the N- and C-termini of ASIC1a and they are all disrupted by extracellular acidosis.

Fig. R8. Both inter- and intra-subunit interactions between the N- and C-termini of ASIC1a are disrupted by extracellular acidosis. (a) Plasmid design for inter- and intra-subunit interactions of ASIC1a. A tandem concatemer which has three ASIC1a subunits

connected as one polypeptide was used. To examine inter-subunit interaction, we tagged a CFP at the N terminus and a YFP at the C terminus (Plasmid 1); to test intra-subunit interaction, we tagged a CFP at the N terminus and inserted a YFP after the first subunit (Plasmid 2). (b) Representative CFP and YFP emission images of CHO cells transfected with Plasmid (1) and Plasmid (2). (c) Acidosis induced dissociation of ASIC1a-CT from its NT both in CHO cells transfected with Plasmid (1) and Plasmid (2). FRET signals determined by the YFP/CFP emission ratios were normalized to that of pH 7.4. The signals were decreased by acidosis (pH 6.0) and recovered upon returning to pH 7.4 SS. n=4–6.

3. For spectral FRET, the exact parameters for spectrum imaging can be added to the methods section. For example, the spectral range captured and used for calculating the efficiency.

Response: We apologize for the insufficient information of spectral FRET analysis. Following the suggestion, we have provided the exact spectral range in the revised manuscript (see pages 27 to 28, lines 595 to 623).

4. Despite the explanation by the authors, it is not so easy to fully digest the observation that the del20 mutant has greatly reduced expression but potent effect on inducing cell death.

Response: We apologize for the inadequate explanation here. We have revised the sentence as “Similarly, the expression levels of the $\Delta 1-20$ mutant were also much lower than that of WT ASIC1a (Fig. 2i). The reduced expression may be directly related to the ability of $\Delta 1-20$ to cause cell death, as cells with high levels of $\Delta 1-20$ expression would have been dead. Alternatively, ASIC1a might also undergo activity (or use)-dependent degradation, like RIPK1². Consistent with the previous finding that ASIC1a-mediated RIPK1 activation also causes RIPK1 degradation², the levels of RIPK1 were also reduced in cells that expressed $\Delta 1-20$ (Fig. 2i)”. Please see page 10, lines 217 to 222.

Reviewer #3 (Remarks to the Author):

This research team has previously shown that ASIC1a can mediate necroptosis induced by acidification, via recruiting the kinase RIPK1 to its C-terminus. This process is independent of any ion currents passing through ASIC1a. Here they test the hypothesis that the intracellular N-terminal part of ASIC1a exerts under baseline conditions an autoinhibitory function by interacting with the C-terminus and preventing the interaction of the C-terminus with RIPK1. In support of this hypothesis, the authors show first that a peptide corresponding to a part of the ASIC1a N terminus prevents acid-induced cell death, and show further that expression of a N-terminally truncated ASIC1a construct (thus a construct lacking the autoinhibitory sequence) induces cell death at physiological pH. An interaction partner of the ASIC1a N terminus, N-ethylmaleimide-sensitive fusion ATPase (NSF) is identified, which interacts with the N-terminus and promotes association of RIPK1 with the C-terminus. To understand the interactions between the

ASIC1a N- and C-termini, the authors construct a structural model of the intracellular ASIC1a parts. They find that a stretch of Glu residues of the N-terminus is close to the RIPK1 interaction sequence present in the C-terminal sequence containing several Lys residues, suggesting electrostatic interactions between the termini. To describe possible changes in their conformation, the authors express ASIC constructs tagged with CFP and YFP at their N- and C-termini and carry out FRET experiments. These experiments indicate a slow decrease of FRET upon acidification, which is interpreted as a separation of the N- and C-termini. Finally, it is shown that a membrane-permeable version of the N-terminal peptide reduces the infarct volume in an ischemic stroke model. ASICs play an important role in the context of ischemic stroke, and their mode of action in this context has been a puzzle for a long time. This study is an interesting follow-up of the previous study, showing clear evidence for a protective role of an intracellular N-terminal ASIC1a sequence.

General comments

1. The FRET experiments are a central part of the study. It is however very difficult to imagine how changes in FRET between ASIC-attached CFP and YFP should represent changes in distance between the N-terminal EEE and the C-terminal KKK motif. First, CFP and YFP are quite big proteins of 35-40 kDa each. According to the description, an ASIC trimer would contain 3 CFP and 3 YFP molecules, all attached to the relatively short ASIC N- and C-termini. The reporter fluorophores are therefore much bigger than the peptide sequences whose movement they should record. How can the authors be sure that changes in FRET report differences in distance between the ASIC subunit N- and C-termini? Smaller fluorophores need to be used for this assay. The EEE motif is located at the very beginning of the ASIC1a N-terminus, while the motif with the K residues is more in the center of the C-terminus. To be able to follow the interaction between these motifs, it would be good to place the fluorophore of the C-terminus close to the RIPK1 interaction motif, and not at its end. Based on the structural model of the intracellular part of the ASIC trimer, it needs to be shown how about these CFP and YFP molecules would be positioned relative to the ASIC channel.

Response: We thank the reviewer for the comment and constructive suggestions. We agree with the reviewer that the reporter fluorophores we used are much bigger than the short ASIC1a NT and CT, which may affect the interaction of NT and CT. Smaller fluorophores are ideal alternatives to solve this problem. We have considered using single molecular FRET or transition metal ion FRET with unnatural amino acid (e.g. L-ANAP) to address this issue. However, these techniques are difficult to implement in a relatively short time period because the targeting sites for the fluorophores can only be determined through trial-and-error. As such, it requires a tremendous amount of time and efforts, as well as some luck, to establish the proper target sites and experimental conditions for the FRET and it is hard to predict when such sites (able to generate FRET) will be found. Nonetheless, FRET using smaller fluorophores for ASIC1a N- and C-termini is on top of our to-do list for the future study.

We would like to emphasize that our main conclusion on the presence of an N- to C- interaction at the cytoplasmic termini of ASIC1a and its dissociation upon acid

stimulation is based on a collective set of experimentation, including studying the necroptotic effects of ASIC1a-NT deletion, the protective effects of the NT peptide on death induced by acidosis and ASIC1a-CT peptide, the protective effects of NSF knockdown, and the Rosetta modelling. Even though the current FRET assay has some inherited caveats, the results are consistent with an acidosis-induced relative movement between ASIC1a-NT and CT, which is in line with our main conclusion.

2. For some of the approaches, the methods and conditions are not sufficiently well (or in some cases not at all) presented. For the approaches, this concerns the modeling and testing of the models, which is not explained, and the constructs for the FRET experiments. Besides providing information on the construction, testing and selection of the models, the quality, number and size of the structural model images in the manuscript needs to be improved, to allow the reader to understand the predictions. Are the CFP and YFP attached to the ends of full length ASIC1a, or to truncated constructs, as Suppl. Fig. 2a might indicate? In the context of the CFP and YFP constructs, the term “concatemer” is used. Does this apply to the CFP-YFP construct, or to any ASIC concatemers? The conditions and equipment for the FRET experiments need to be described.

Response: We again appreciate the reviewer’s comments and suggestions. We followed the suggestions and provided more detailed descriptions on the conditions and equipment used for the FRET experiments (see pages 27 to 28, lines 591 to 623). We have also updated the structural model in Figs. 2a and 6a and numbered the key amino acid sequences in Fig. 6a. As shown in Supplementary Figure 2a, the CFP and YFP were attached to the NT and CT ends of full-length ASIC1a, respectively. In Fig. 2c (now 2d in the revised ms), the CFP-YFP concatemer contains only CFP-YFP with no ASIC1a included.

Indicate how the bands of Western blots were quantified. For the quantitative analysis of band intensities (Figs. 2i, 6f, 6g) indicate to which condition the signal intensities are normalized.

Response: We thank the reviewer for the comments. We have indicated these details in the revised manuscript. The band intensities of Western blots were quantified by ImageJ with background subtraction (see pages 25 to 26, lines 557 to 568). The data in Fig. 2j (previously 2i) are normalized to RIPK1/ASIC1a of WT group in pH=7.4, and the data in Fig. 6f and 6g are normalized to NSF/ASIC1a and RIPK1/ASIC1a of WT group in pH=7.4, respectively. We have added such information to the corresponding figure legends.

For the following experiments, the conditions should be better defined: Fig. 2d-e: what was the pH of incubation? Fig. 6h, indicate the Nec-1 and NT1-20 and NT1-20E/A peptide concentrations used.

Response: Thanks for the comment. We followed the suggestion and provided the information in the revised manuscript: the pH of incubation in Fig. 2e and 2f is 7.4 (see

page 36, line 873), and the concentrations of Nec-1, NT₁₋₂₀ and NT₁₋₂₀^{E/A} in Fig. 6h were 20 μM, 10 μM, and 10 μM, respectively (see page 40, lines 963 to 965).

3. It is reported in this study that some of the truncation mutants, and the EEE-AAA mutant show very low expression. In the context of the elucidation of the mechanism of the involvement of ASIC1a in necroptosis, it is important to know whether the channels need to be present at the plasma membrane in order to mediate necroptosis. It is therefore critical to provide data on the cell surface expression of these mutants.

Response: We thank the reviewer for the comment. We have shown that deletion of aa1-20 (Δ1-20) decreased proton-evoked currents of ASIC1a and reduced the total expression of ASIC1a (Fig. 2g-j). To address the effects of E/A mutation, we have measured the whole-cell currents, as well as the total and surface expression levels of ASIC1a E/A mutants, in comparison with the wild type channel. The data from these new experiments show that E/A mutation caused decreases in the current and protein expression (both total and surface) of ASIC1a (Fig. R9). We suspect that the increased conformational signaling due to these mutations rendered the cells with high expression of the mutant protein dead and/or the functionality of the mutant ASIC1a proteins caused themselves to be degraded. Please see page 15, lines 325 to 329 for the revised text. Please also refer to revised Supplementary Figures 4c-e and the legends for details (see page 43, lines 1020 to 1024).

*Fig. R9. E/A mutation caused decreases in the currents and protein expression (both total and surface) of ASIC1a. (a) Representative pH 6.0-induced current traces at -60 mV of CHO cells transfected with WT and E/A mutant ASIC1a. (b) summary data for peak currents of WT and E/A mutant ASIC1a expressed in CHO cells. n=16–19, *p<0.05 vs. WT. (c) E/A mutation decreased the total expression of ASIC1a compared with WT, but not the surface expression (Surface/Total ratio).*

Specific points

1. Are RIPK1 and NSF endogenously expressed in CHO cells?

Response: Yes, based on our data, RIPK1 and NSF are endogenously expressed in CHO cells (Fig. 2i and Fig. 4c).

2. The observation that the NT1-20 peptide has a protective effect at 5 and 10 microM, but not at 20 microM, is somewhat confusing. How could the decrease in protection at higher peptide concentrations be explained? This should be discussed in the manuscript.

Response: We thank the reviewer for pointing out this issue, which was also raised by Reviewers #1 and #2. Please refer to our response to reviewer #1's Minor comments 2) and 3). Briefly, 20 μ M NT₁₋₂₀ lost the protection because of the toxicity from the TAT tag (Fig. R2), which was used to facilitate cell penetration. Future work should focus on identifying alternative sequences to facilitate cell penetration.

3. How is it explained that PcTx1 prevents in some conditions the ASIC1a-mediated necroptosis? By shifting the pH dependence of inactivation, PcTx1 affects ASIC opening. How would this affect a signaling that does not depend on ion permeation? Demonstration of prevention or inhibition of the conformational changes in the intracellular ASIC parts by PcTx1 would be a strong argument in favor of the proposed mechanisms of necroptosis.

Response: We thank the reviewer for the comment and suggestion. We have discussed the potential mechanisms underlying the protective effect of PcTX1 against acid-induced necroptosis in the previous paper (Wang et al., eLife 2015)⁵. We proposed that extracellular protons cause at least two steps in conformational changes of ASIC1a: one that frees the CT to activate RIPK1 and the other that mediates channel gating. The two steps may be linked and thus both can be inhibited by PcTX1. To provide direct evidence for this hypothesis, we performed additional experiments testing the effect of PcTX1 on acid-induced NT-CT dissociation of ASIC1a by FRET. As expected, PcTX1 significantly reduced the low pH-induced change in FRET, indicative of a block in the dissociation of NT from CT (Fig. R10, also Fig. 2b-c in the revised ms). We have updated this part in the revised manuscript accordingly (see page 9, lines 177 to 179).

Fig. R10. The effects of PcTX1 and NSF shRNA on acid-induced NT-CT dissociation of ASIC1a by FRET. (a) Both PcTX1 and NSF shRNA significantly reduced acid-induced change in FRET, indicative of a block in the dissociation of NT from CT. (b) Summary of the magnitude of fluorescence ratio change.

4. Page 13, bottom, “Therefore, disrupting...” Since the NSF binding site is on the N-terminal, it is also quite likely that this mutation could also interfere with NSF binding. How can the authors exclude such a possibility?

Response: We thank the reviewer for bringing up this question. We agree with the reviewer that there is a possibility that the E/A mutation may interfere with the interaction between ASIC-NT and NSF. However, the co-IP data in Fig. 6e-f showed an increase, rather than a decrease, in the association between ASIC1a-E/A and NSF even at neutral pH. This indicates that the glutamates are mainly involved in the electrostatic interaction with the C-terminal lysine residues. With the E → A substitution, the N terminus is dissociated from the CT and therefore more ready to recruit NSF, explaining the increased co-IP results. Accordingly, we consider that the glutamates may not contribute to the interaction between ASIC-NT and NSF.

5. There is clear evidence for a role of NSF in contributing to necroptosis. It seems however that this effect could be independent of the N- and C-terminus interaction. The authors should clearly develop their arguments for a role of NSF in inhibiting the interaction between the ASIC1a N- and C-termini, and, if this is not possible, present their conclusion (“Plausibly, by binding to ASIC1a NT, NSF helps keep the CT death motif free to interact with RIPK1, ...”) as hypothesis.

Response: We thank the reviewer for the comment and suggestions. To address this question, we performed additional FRET experiments in NSF shRNA-transfected cells. As expected, the knockdown of NSF impaired the acid-induced dissociation of ASIC1a NT from its CT, as shown by the shallower decrease in FRET than the blank control (Fig. R10, also Fig. 5e of the revised ms). The data suggest that NSF is likely involved in stabilizing the dissociation between ASIC1a-NT and CT under acidosis. We have updated this part in the revised manuscript accordingly (see page 14, lines 298 to 301). Please refer to revised Fig. 5e-f and the legend for details (see page 39, lines 928 to 932).

6. P.16, lower paragraph, “Unfortunately, all these structures lack the..”. It is not true that all these constructs are non-functional. Some of them were shown to be functional.

Response: We thank the reviewer for the comment, which was also raised by Reviewer #1. Please refer to our response to Reviewer #1’s Minor comments 9). We have revised the statement (see page 18, line 390).

7. In the legend to Figs. 2d-e it is indicated that the PI staining was carried out after 24 h. Does this mean 24h after the transfection? Please clarify. The inhibitors were added 1h before the PI staining in these experiments. Assuming that the ASIC constructs were expressed a few hours after transfection, how can it be explained that exposure to the inhibitors in the last hour was sufficient to prevent cell death?

Response: We apologize for the misleading data presentation. We have double checked our experiment protocol and revised the figure legend of Fig. 2e-f as follows: “(e, f) Representative images (e) and summary (f) of PI staining of CHO cells transfected with

*WT ASIC1a and its HIF and NT truncation mutants. Nec-1, peptide NT₁₋₂₀, or PcTX1 was added to NT truncated ASIC1a cells immediately after transfection, and PI staining was carried out 24 h after transfection. The deletion of aa1-20 (Δ 1-20) induced cell death, which was prevented by Nec-1 and peptide NT₁₋₂₀, but not PcTX1. n>200 CHO cells counted for each. ***p<0.001, vs. WT; #### p<0.001 vs. Δ 1-20.” Please see pages 36 to 37, lines 871 to 878.*

8. Fig. 2g, the current amplitudes need to be indicated as absolute current amplitude or current densities, not as normalized values.

Response: We accepted the reviewer’s comment and have revised Fig. 2g with current amplitude.

References

- 1 Gwiazda, K., Bonifacio, G., Vullo, S. & Kellenberger, S. Extracellular subunit interactions control transitions between functional states of acid-sensing ion channel 1a. *J Biol Chem* **290**, 17956-17966, doi:10.1074/jbc.M115.641688 (2015).
- 2 Bozzelli, P. L. *et al.* HIV-1 Tat promotes astrocytic release of CCL2 through MMP/PAR-1 signaling. *Glia* **67**, 1719-1729, doi:10.1002/glia.23642 (2019).
- 3 Toborek, M. *et al.* HIV-Tat protein induces oxidative and inflammatory pathways in brain endothelium. *J Neurochem* **84**, 169-179, doi:10.1046/j.1471-4159.2003.01543.x (2003).
- 4 Takanishi, C. L., Bykova, E. A., Cheng, W. & Zheng, J. GFP-based FRET analysis in live cells. *Brain Res* **1091**, 132-139, doi:10.1016/j.brainres.2006.01.119 (2006).
- 5 Wang, Y. Z. *et al.* Tissue acidosis induces neuronal necroptosis via ASIC1a channel independent of its ionic conduction. *Elife* **4**, doi:10.7554/eLife.05682 (2015).

Reviewers' Comments:

Reviewer #1:

Remarks to the Author:

I appreciate that the authors made considerable efforts to address the comments of the referees and to improve their manuscript. I have no further comments.

I congratulate the authors to this nice study.

Reviewer #2:

Remarks to the Author:

All my comments are addressed. The lack of effect by acidosis on CFP-YFP dimer is a little unexpected. This result, however, serves as a good control for the CFP-ASIC-YFP construct.

Reviewer #3:

Remarks to the Author:

The authors have carried out additional experiments, which in several cases further supported the conclusions. The additional conditions tested with the FRET analysis showed the results that were expected, thus they suggest that the CFP/YFP pair may indeed measure distance changes between the ASIC1a N- and C-terminus. It remains however that these reporters are big proteins, much bigger than the cytoplasmic N- and C-termini of ASIC1a whose movement they should report. This is a limit of the study, and the authors need to mention and discuss this limit in the text, either in the results or discussion section.

I found the additional experiments addressing intracellular pH changes, shown in Fig. R1, intriguing and interesting. It seems however that this additional information, except for Fig R1d, is not shown in the revised manuscript. These data should be included and discussed in the revised manuscript.

REVIEWERS' COMMENTS:

Reviewer #1 (Remarks to the Author):

I appreciate that the authors made considerable efforts to address the comments of the referees and to improve their manuscript. I have no further comments.

I congratulate the authors to this nice study.

Response: We greatly appreciate the reviewer for the positive comment.

Reviewer #2 (Remarks to the Author):

All my comments are addressed. The lack of effect by acidosis on CFP-YFP dimer is a little unexpected. This result, however, serves as a good control for the CFP-ASIC-YFP construct.

Response: We greatly appreciate the reviewer for the positive comment.

Reviewer #3 (Remarks to the Author):

The authors have carried out additional experiments, which in several cases further supported the conclusions. The additional conditions tested with the FRET analysis showed the results that were expected, thus they suggest that the CFP/YFP pair may indeed measure distance changes between the ASIC1a N- and C-terminus. It remains however that these reporters are big proteins, much bigger than the cytoplasmic N- and C-termini of ASIC1a whose movement they should report. This is a limit of the study, and the authors need to mention and discuss this limit in the text, either in the results or discussion.

I found the additional experiments addressing intracellular pH changes, shown in Fig. R1, intriguing and interesting. It seems however that this additional information, except for Fig R1d, is not shown in the revised manuscript. These data should be included and discussed in the revised manuscript.

Response: We highly appreciate the reviewer for the insightful and helpful suggestions that allowed us to improve the manuscript. As suggested, we have included the limitation of the reporter fluorophores we used in FRET analysis (see page 19, lines 388 to 391) and the Fig. R1 (now Supplementary Figure 3) in the new manuscript.